# Calibrated Stackelberg Games: Learning Optimal Commitments Against Calibrated Agents

**Nika Haghtalab**[1], **Chara Podimata**[2], and **Kunhe Yang**[1]

[1]University of California, Berkeley, {nika,kunheyang}@berkeley.edu
[2]MIT & Archimedes, podimata@mit.edu

## Abstract

In this paper, we introduce a generalization of the standard Stackelberg Games (SGs) framework: *Calibrated Stackelberg Games (CSGs)*. In CSGs, a principal repeatedly interacts with an agent who (contrary to standard SGs) does not have direct access to the principal's action but instead best-responds to *calibrated forecasts* about it. CSG is a powerful modeling tool that goes beyond assuming that agents use ad hoc and highly specified algorithms for interacting in strategic settings and thus more robustly addresses real-life applications that SGs were originally intended to capture. Along with CSGs, we also introduce a stronger notion of calibration, termed *adaptive calibration*, that provides fine-grained any-time calibration guarantees against adversarial sequences. We give a general approach for obtaining adaptive calibration algorithms and specialize them for finite CSGs. In our main technical result, we show that in CSGs, the principal can achieve utility that converges to the optimum Stackelberg value of the game both in *finite* and *continuous* settings, and that no higher utility is achievable. Two prominent and immediate applications of our results are the settings of learning in Stackelberg Security Games and strategic classification, both against *calibrated* agents.

## 1 Introduction

Stackelberg games (SGs) are a canonical model for strategic principal-agent interactions, considering a principal (or "leader") that commits to a strategy $\mathbf{h}$ and an agent (or "follower") who observes this strategy and best respond by taking action $\mathrm{BR}(\mathbf{h})$. These games are inspired by real-world applications such as economic policy design (where a tax policymaker establishes rules for triggering audits before taxes are filed), defense (where a principal allocates security resources to high-risk targets before vulnerabilities are exploited) and many more, see e.g., [7, 22, 49, 35, 21, 15, 17]. By anticipating the agent's best-response, a principal who knows the agent's payoff function can calculate the *optimal Stackelberg strategy* guaranteeing her utility $V^\star$. In recent years, *repeated* SGs have gained popularity in addressing settings where the agent's payoff function is *unknown* to the principal. In this setting, the principal, who can only observe the agents' actions, aims to deploy a sequence of strategies $\mathbf{h}_1, \ldots, \mathbf{h}_T$ over $T$ rounds whose average payoff is at least as good as $V^\star$, i.e., the value of her optimal strategy had she known the agent's payoffs in advance.

Despite the original intent, repeated SGs are often studied under strict assumptions on the agent's knowledge and algorithmic behavior. Examples include requiring the agent to best respond per round using $y_t = \mathrm{BR}(\mathbf{h}_t)$ [7, 21], necessitating the agent to precisely know the principal's strategy at all times (e.g., the attacker must anticipate the exact probabilistic allocation of the defender's security resources), or employing one of many online optimization algorithms whose every detail (down to the learning step size) can significantly impact the principal's utility [51].

In this paper, instead of working with such restrictive and often unrealistic assumptions on the agent's knowledge and behavior, we build on foundational decision theoretic concepts, such as *forecasts* and *calibration* [19, 29, 30]. In practice, while agents may not observe the principal's true strategies $\mathbf{h}_t$, they can form *calibrated forecasts* — a notion of consistency in beliefs about $\mathbf{h}_t$ — to which they then best respond. Indeed, such a decision-theoretic perspective on game dynamics led to seminal results on converging to correlated and Nash equilibria in simultaneous multi-player games [29, 38]. Our work brings the perspective of calibrated forecasts to principal-agent games. We introduce *Calibrated Stackelberg Games (CSG)*—a class that is more general than standard SGs— and ask:

**Q1.** *What characterizes principal's optimal utility in CSGs?*

**Q2.** *Are there natural forecasting algorithms for the agent that satisfy calibration?*

**Our Contributions.** We answer both questions completely. For **Q1**, we show that the principal's optimal utility converges *exactly* to $V^\star$. For **Q2**, we give a general approach for obtaining a fine-grained any-time notion of calibration *of independent interest* and further specializing it to games.

Before we delve into the details of our contributions, we highlight two key aspects of our results. First, *calibration* is a common property of forecasting procedures shared by many algorithms, not any one particular algorithm defining the agent's behavior. Despite not constraining the agent to any particular algorithm, our answer to **Q1** shows that the principal can meaningfully converge to $V^\star$, which is the value she could have achieved in a single-shot game had she known the agent's utility. Second, our definition and results immediately apply to two important Stackelberg settings; Stackelberg Security Games [49, 7, 33] and strategic classification [21]. As such, we obtain the first results for learning against calibrated agents in these settings too.

Our work contributes two concepts of independent interest (Section 2): first, *CSGs* that directly generalize the standard model of repeated SGs, and second, a notion of calibration, termed *adaptive calibration*. This notion which draws inspiration from *adaptive regret bounds* in online learning, provides fine-grained calibration guarantees for adversarial sequences.

Beyond the introduction of these models, we address an important property of CSGs in our answer to **Q1**. We show that the principal's optimal utility in CSGs converges to $V^\star$, nothing more or less, in games with *finite* (Section 3) or *continuous* (Section 5) actions spaces. Note that $V^\star$ is a benchmark that gives both players more power: the principal knows the agent's utility and the agent observes the principal's strategy. We find it somewhat surprising then that the optimal achievable principal utility in CSGs, in which both players work with significantly less knowledge, converges to $V^\star$ exactly.

As for our newly introduced notion of adaptive calibration (Section 4), we provide an answer to **Q2** by giving a general approach for creating adaptively calibrated forecasting algorithms. This shows that adaptive calibration is not just an abstract notion, rather, it is a natural property with deep roots in the theory of online learning. Indeed, to obtain these results we draw inspirations from recent advances in the multicalibration literature [34] regarding simulating no-regret and best-response dynamics and the sleeping experts problem setting which has been a staple of the online learning literature [8, 31]. Furthermore, we specialize our approach for strategic settings by showing how standard calibration concepts (such as the "binning function") can be adapted to account for the agent's best-responses.

## 1.1 Related work

**Repeated Stackelberg games.** Learning optimal Stackelberg strategies has been studied in the offline [17] and the online setting, where only instantaneous best-responses are observable (i.e., no access to a best-response oracle). Key applications include Stackelberg Security Games (e.g., [12, 7, 47, 50]) and strategic classification (e.g., [21, 15, 3, 4]). Another line of work treats repeated games as an extensive form game and studies optimal strategies for infinite [52] or finite [16] horizons. Other works consider learning in the presence of non-myopic agents that best respond by maximizing discounted utilities [5, 33, 2]. The main distinction to our work is that in our setting, the agents have only calibrated forecasts regarding the principal's strategies (rather than full knowledge of them).

**Stackelberg games beyond best responses.** Recent works have studied variants of repeated Stackelberg games with different agent strategic behaviors beyond best responding. One example is no-regret learning, in which previous works have extensively investigated the relationship between the principal's cumulative utility and the single-shot Stackelberg value against agents that use mean-based learning algorithms [13], gradient descent [24, 25], no-external regret algorithms [13, 20, 51],

no-internal/swap regret algorithms [20, 45], and no-counterfactual internal regret algorithms [14]. Another research direction assumes agents approximately best respond due to uncertainty in the principal's strategy [11, 6, 46] or their own [43, 39, 40] and study *robust Stackelberg equilibria* [48, 32]. Most of the works here assume that the principal knows the agent's utility function with the exception of [51]. Core differences to our framework are that (1) we work in an online learning setting where the principal has to learn the agent's utility function from their responses; (2) we do not assume a specific agent algorithm but focus on properties of agent beliefs that are shared by many algorithms.

**Calibration and application in games.** The study of calibration, introduced by Dawid [19], dates back to seminal work by Foster and Vohra [30] and Hart [36] that showed the existence of asymptotic online calibration against any adversarial sequence. Applying calibration to game dynamics, Foster and Vohra [29] introduced the concept of *calibrated learning*, which refers to a player best responding to calibrated forecasts of others' actions. They demonstrated that the game dynamics of all players performing calibrated learning converge to the set of correlated equilibria. This is complemented by the results of [38, 27, 28] that show *smooth and continuous* variants of calibrated learning dynamics converge to Nash equilibrium. Our work differs from the above by studying game dynamics that converge to a Stackelberg equilibrium, where only the follower (agent) performs calibrated learning.

**Adaptivity and sleeping experts.** The notion of adaptive calibration introduced in Section 2 is related to the study of adaptivity of regret bounds in online learning [44, 18, 37]. Our design of adaptively calibrated forecasting algorithms builds on the *multi-objective learning* perspective of online (multi-)calibration [41, 34] and the powerful tool of *sleeping experts* [8, 31, 44] which has proven useful in various applications such as fairness [9].

## 2 Model & preliminaries

We begin this section with some basic definitions about forecasts, calibration, and games, and then introduce the class of games that we study; *Calibrated Stackelberg Games* (CSGs).

**Adaptively Calibrated Forecasts.** We use $A$ to denote the space of outcomes and $C \supseteq A$ to denote the space of forecasts. A (stochastic) forecasting procedure $\sigma$ is an online procedure that takes any adversarial sequence of outcomes $\mathbf{h}_t \in A$ for $t \in [T]$, and on round $t$ outputs (possibly at random) forecast $\mathbf{p}_t \in C$ solely based on outcomes and forecasts $\mathbf{h}_\tau, \mathbf{p}_\tau$, for $\tau \in [t-1]$. To define calibrated forecasts, let us first introduce the notion of *binning functions*.

**Definition 2.1** (Binning [28]). *We call a set $\Pi = \{w_i\}_{i \in [n]}$ a* binning function, *if each $w_i : C \to [0, 1]$ maps forecasts to real values in $[0, 1]$, and for all $\mathbf{p} \in C$ we have $\sum_{i \in [n]} w_i(\mathbf{p}) = 1$.*

With the above binning functions, we define the adaptive calibration error with respect to $\Pi$ as follows. At a high level, conditioned on any bin, the calibration error measures the difference between the expected forecasts that fall in that bin and the corresponding expected outcome.

**Definition 2.2** ($\Pi$-Adaptive Calibration Error). *For any time interval $[s, t]$, let $\mathbf{p}_{s:t}$ be the sequence of forecasts and $\mathbf{h}_{s:t}$ be the sequence of outcomes. For a given binning $\Pi = \{w_i\}_{i \in [n]}$ with size $n$, and $\forall i \in [n]$, define the $\Pi$-adaptive calibration error as*

$$\text{CalErr}_i \left( \mathbf{h}_{s:t}, \mathbf{p}_{s:t} \right) \triangleq \frac{n_{[s,t]}(i)}{t - s} \cdot \left\| \bar{\mathbf{p}}_{[s,t]}(i) - \bar{\mathbf{h}}_{[s,t]}(i) \right\|_\infty, \tag{1}$$

*where during interval $[s, t]$, $n_{[s,t]}(i) \triangleq \sum_{\tau=s}^{t} w_i(\mathbf{p}_\tau)$ is the effective number of times that the forecast belongs to bin $i$ (i.e., bin $i$ is activated), $\bar{\mathbf{p}}_{[s,t]}(i) \triangleq \sum_{\tau=s}^{t} \frac{w_i(\mathbf{p}_\tau)}{n_{[s,t]}(i)} \cdot \mathbf{p}_\tau$ is the expected forecast that activates bin $i$, $\bar{\mathbf{h}}_{[s,t]}(i) \triangleq \sum_{\tau=s}^{t} \frac{w_i(\mathbf{p}_\tau)}{n_{[s,t]}(i)} \cdot \mathbf{h}_\tau$ is the expected outcomes corresponding to bin $i$.*

We say that a forecasting procedure is adaptively calibrated if it achieves vanishing calibration error on any adversarial sequence of outcomes and any sub-interval of time steps.

**Definition 2.3** (($\varepsilon, \Pi$)-Adaptively Calibrated Forecasts). *A forecasting procedure $\sigma$ is $\varepsilon$-adaptively calibrated to binning $\Pi = \{w_i\}_{i \in [n]}$ with rate $r_\delta(\cdot) \in o(1)$, if for all adversarial sequences of actions $\mathbf{h}_1, \cdots, \mathbf{h}_T$, where $\mathbf{h}_t \in A$, $\sigma$ outputs forecasts $\mathbf{p}_t \in C$ for $t \in [T]$ such that with probability at least $1 - \delta$, we have that $\forall s, t$ such that $1 \leq s < t \leq T$, and $\forall i \in [n]$:*

$$\text{CalErr}_i \left( \mathbf{h}_{s:t}, \mathbf{p}_{s:t} \right) \leq r_\delta(t - s) + \varepsilon.$$

We remark that without adaptivity (i.e., for $s = 1$ and $t = T$), Definition 2.2 is weaker than the standard definition of calibration (e.g., [30], listed for completeness in Appendix A) in two ways: (1) standard calibration takes each prediction $\mathbf{p} \in C$ as an independent bin, thus having infinitely many binning functions: $w_{\mathbf{p}}(\cdot) = \delta_{\mathbf{p}}(\cdot)$. Instead, we only require calibration with respect to the predefined binning $\Pi$ which only contains a finite number of binning functions; (2) standard calibration cares about the summation over calibration error across bins, but we only consider the maximum error.

**Stackelberg Games.** A *Stackelberg game* is defined as the tuple $(\mathcal{A}_P, \mathcal{A}_A, U_P, U_A)$, where $\mathcal{A}_P$ and $\mathcal{A}_A$ are the principal and the agent action spaces respectively, and $U_P : \mathcal{A}_P \times \mathcal{A}_A \to \mathbb{R}_+$ and $U_A : \mathcal{A}_P \times \mathcal{A}_A \to \mathbb{R}_+$ are the principal and the agent utility functions respectively. For ease of exposition, we work with *finite* Stackelberg games (i.e., $|\mathcal{A}_P| = m$ and $|\mathcal{A}_A| = k$) and generalize our results to continuous games in Section 5. When the principal plays action $x \in \mathcal{A}_P$ and the agent plays action $y \in \mathcal{A}_A$, then the principal and the agent receive utilities $U_P(x, y)$ and $U_A(x, y)$ respectively. We also define the principal's *strategy space* as the simplex over actions: $\mathcal{H}_P = \Delta(\mathcal{A}_P)$. For a strategy $\mathbf{h} \in \mathcal{H}_P$, we oftentimes abuse notation slightly and write $U_P(\mathbf{h}, y) := \mathbb{E}_{x \sim \mathbf{h}}[U_P(x, y)]$.

*Repeated Stackelberg games* capture the *repeated* interaction between a principal and an agent over $T$ rounds. What distinguishes Stackelberg games from other types of games is the inter-temporal relationship between the principal's action/strategy and the agent's response; specifically, the principal first commits to a strategy $\mathbf{h}_t \in \mathcal{H}_P$ and the agent subsequently *best-responds* to it with $y_t \in \mathcal{A}_A$. Let $\mathbf{p}_t \in \mathcal{F}_P = \mathcal{H}_P$ be the agent's *belief* regarding the principal's strategy at round $t$. In standard Stackelberg games: $\mathbf{p}_t = \mathbf{h}_t$, i.e., the agent has full knowledge of the principal's strategy. In this paper, we consider games where the agent does not in general know $\mathbf{h}_t$ when playing, but they only best-respond according to their belief $\mathbf{p}_t$. The agent's *best-response* to *belief* $\mathbf{p}_t$ according to her underlying utility function $U_A$ is action $y_t \in \mathcal{A}_A$ such that

$$y_t \in \mathrm{BR}(\mathbf{p}_t) \quad \text{where} \quad \mathrm{BR}(\mathbf{p}_t) = \operatorname*{argmax}_{y \in \mathcal{A}_A} \mathbb{E}_{x \sim \mathbf{p}_t}[U_A(x, y)]. \tag{2}$$

We often overload notation and write $U_A(\mathbf{p}, y) := \mathbb{E}_{x \sim \mathbf{p}}[U_A(x, y)]$. Note that from Equation (2), the best-responses to $\mathbf{p}_t$ form *a set*. If this set is not a singleton, we use either a *deterministic* or a *randomized tie-breaking* rule. For the *deterministic* tie-breaking rule, the agent breaks ties according to a predefined preference rule $\succ$ over the set of actions $\mathcal{A}_A$. For the *randomized* tie-breaking rule, the agent chooses $y_t$ by sampling from the set $\mathrm{BR}(\mathbf{p}_t)$ uniformly at random, i.e., $y_t \sim \mathrm{Unif}(\mathrm{BR}(\mathbf{p}_t))$.

The *Stackelberg value* of the game is the principal's optimal utility when the agent best responds:

$$V^\star = \max_{\mathbf{h}^\star \in \mathcal{H}_P} \max_{y^\star \in \mathrm{BR}(\mathbf{h}^\star)} U_P(\mathbf{h}^\star, y^\star).$$

In the above definition $\mathbf{h}^\star$ is referred to as the *principal's optimal strategy*.

For an agent's action $y \in \mathcal{A}_A$, we define the corresponding *best-response polytope* $P_y$ as the set of all of the agent's beliefs that induce $y$ as the agent's best-response, i.e., $P_y = \{\mathbf{p} \in \mathcal{F}_P : y \in \mathrm{BR}(\mathbf{p})\}$. We make the following standard assumption, which intuitively means that there are sufficiently many strategies that induce $y^\star$ as the agent's best-response.

**Assumption 2.4** (Regularity). *The principal's optimal strategy* $\mathbf{h}^\star \in \Delta(\mathcal{A}_P)$ *and the agent's optimal action* $y^\star \in \mathrm{BR}(\mathbf{h}^\star)$ *satisfy a regularity condition:* $P_{y^\star}$ *contains an* $\ell_2$ *ball of radius* $\eta > 0$.

---

**Interaction Protocol for Calibrated Stackelberg Games (CSGs)**

(1) The principal plays strategy $\mathbf{h}_t \in \mathcal{H}_P$.

(2) The agent *without observing* $\mathbf{h}_t$ forms a *calibrated prediction* $\mathbf{p}_t$ (Def. 2.5) about $\mathbf{h}_t$.

(3) The agent best responds to $\mathbf{p}_t$ by playing $y_t \in \mathrm{BR}(\mathbf{p}_t)$ (+ tie-breaking).

(4) The principal observes $y_t$ and experiences utility $U_P(\mathbf{h}_t, y_t)$.

(5) The agent observes $\mathbf{h}_t$, or an action sampled from $\mathbf{h}_t$. [1]

---

Figure 1: Principal-Agent Interaction Protocol in a Round of a CSG

**Calibrated Stackelberg Games.** In CSGs (see Figure 1 for the principal-agent interaction protocol[1]), the agent forms $(\varepsilon, \Pi)$-adaptively calibrated forecasts as their beliefs $\mathbf{p}_t$ regarding $\mathbf{h}_t$.

---

[1]If the agent observes action $x_t \sim \mathbf{h}_t$ instead of the mixed strategy $\mathbf{h}_t$, then they can still calibrate to the sequence of $\mathbf{h}_t$ with an additional (vanishing) error term that comes from concentration inequalities.

We first define binning functions that are especially appropriate for forecasts in games. In CSGs, we define $\Pi$ based on whether $i$ is a best-response to the input calibrated forecast, i.e., $\forall \mathbf{p} \in \mathcal{F}_P$:

$$w_i(\mathbf{p}) = \mathbf{1}\{i \in \mathrm{BR}(\mathbf{p}), i \succ j, \forall j \neq i\} \qquad \text{(for the \textit{deterministic} tie-breaking)}$$

$$w_i(\mathbf{p}) = \frac{\mathbf{1}\{i \in \mathrm{BR}(\mathbf{p})\}}{|\mathrm{BR}(\mathbf{p})|} \qquad \text{(for the \textit{randomized} tie-breaking)}$$

Note that both binning functions meet the conditions of Definition 2.1. Applying Definition 2.3 for calibrated agent forecasts in CSGs we have the following:

**Definition 2.5** ($\varepsilon$-Adaptively Calibrated Agent for CSGs). *The agent is called $\varepsilon$-adaptively calibrated with rate $r_\delta(\cdot) \in o(1)$, if for any sequence of principal strategies $\mathbf{h}_1, \cdots, \mathbf{h}_T \in \mathcal{H}_P$ the agent takes a sequence of actions $y_1, \ldots, y_T$ that satisfy the following requirements: 1) there is a sequence of forecasts $\mathbf{p}_t \in \mathcal{F}_P$ for $t \in [T]$, such that $y_t \in \mathrm{BR}(\mathbf{p}_t)$, and 2) forecasts $\mathbf{p}_1, \ldots, \mathbf{p}_T$ are $\varepsilon$-calibrated for binning $\Pi$ with rate $r_\delta(\cdot)$ with respect to the principal's strategies $\mathbf{h}_1, \cdots, \mathbf{h}_T$.*

We next review the fundamental constructs from Equation (1) and their intuitive meaning in this setting. $n_{[s,t]}(i) \triangleq \sum_{\tau \in [s,t]} w_i(\mathbf{p}_\tau)$ is now the expected number of times that the forecast has induced action $i$ from the agent as their best response during interval $[s,t]$, $\bar{\mathbf{p}}_{[s,t]}(i) \triangleq \sum_{\tau \in [s,t]} w_i(\mathbf{p}_\tau) \cdot \mathbf{p}_\tau / n_{[s,t]}(i)$ is the expected forecast that induces action $i$ from the agent as their best response during interval $[s,t]$, and $\bar{\mathbf{h}}_{[s,t]}(i) \triangleq \sum_{\tau \in [s,t]} w_i(\mathbf{p}_\tau) \cdot \mathbf{h}_\tau / n_{[s,t]}(i)$ is the expected principal strategy that induces action $i$ from the agent as their best response during interval $[s,t]$. The requirement for an agent to be calibrated is quite mild, as the forecasts are binned only according to the best-response they induce.

## 3 Principal's learning algorithms

In this section (see Appendix C for full proofs and convergence rates), we study the relationship between the principal's Stackelberg value $V^\star$ and the best utility the principal can obtain from learning to play a sequence of strategies $\{\mathbf{h}_t\}_{t \in [T]}$ against calibrated agents, i.e., $\frac{1}{T} \sum_{t \in [T]} U_P(\mathbf{h}_t, y_t)$. The relationship between $V^\star$ and $\frac{1}{T} \sum_{t \in [T]} U_P(\mathbf{h}_t, y_t)$ is not a priori clear. In the case of calibrated forecasts, the agents do not know the exact $\mathbf{h}_t$ when they choose their response. Instead, they base their decisions on the history of the principal's strategies so far. A principal then may be able to create historical patterns that lead the agents to worse actions, thus obtaining better utility himself. Indeed, several works have shown how historical patterns can afford the principal much better utility than $V^\star$ when the agents are no-regret [13, 20]. Surprisingly, we show that this is not the case when the agents are calibrated; $\sum_{t \in [T]} U_P(\mathbf{h}_t, y_t)$ is upper bounded by $TV^\star$ and a term that is sublinear in $T$ and depends on the calibration parameters.[2]

**Theorem 3.1.** *Assume that the agent is $(\varepsilon, \Pi)$-as calibrated with rate $r_\delta(\cdot)$ and negligible $\varepsilon$. Then, for any sequence $\{\mathbf{h}_t\}_{t \in [T]}$ for the principal's strategies in a CSG, with probability at least $1 - 2\delta$, the principal's utility is upper bounded as: $\lim_{T \to \infty} \frac{1}{T} \sum_{t \in [T]} U_P(\mathbf{h}_t, y_t) \leq V^\star$.*

*Proof sketch.* We sketch the proof for the deterministic tie-breaking, as the randomized one needs just an application of Azuma-Hoeffding. We first rewrite $\sum_{t \in [T]} U_P(\mathbf{h}_t, y_t)$ partitioned in the principal's utility for each round-specific forecast that induces action $i$ as the best-response from the agent, for all actions $i \in \mathcal{A}_A$: $\sum_{t \in [T]} U_P(\mathbf{h}_t, y_t) = \sum_{i \in \mathcal{A}_A} \sum_{t \in [T]} w_i(\mathbf{p}_t) \cdot U_P(\mathbf{h}_t, i)$. This equivalence holds because each $\mathbf{p}_t$ maps to a *single* best response $i \in \mathcal{A}_A$ (deterministic tie-breaking). Because of the linearity of the principal's reward in the principal's strategy: $\sum_{i \in \mathcal{A}_A} \sum_{t \in [T]} w_i(\mathbf{p}_t) \cdot U_P(\mathbf{h}_t, i) = \sum_{i \in \mathcal{A}_A} n_T(i) \cdot U_P(\bar{\mathbf{h}}_T, i)$, where $n_T(i) = n_{[0,T]}(i)$. Adding and subtracting $U_P(\bar{\mathbf{p}}_T, i)$ from the above, we now need to bound the quantity: $\sum_{i \in \mathcal{A}_A} n_T(i)(U_P(\bar{\mathbf{p}}_T(i), i) + \langle U_P(\cdot, i), \bar{\mathbf{h}}_T(i) - \bar{\mathbf{p}}_T(i) \rangle)$. The first term is upper bounded by $V^\star T$; note that $i \in \mathrm{BR}(\bar{\mathbf{p}}_T)$ (and $V^\star = \max_{\mathbf{h}} \max_{y \in \mathrm{BR}(\mathbf{h})} U_P(\mathbf{h}, y)$) since $\mathbf{p}_t(i) \in P_i$ and hence that should also be true for the average of $\mathbf{p}_t(i)$ over $t$ rounds. The second summand is bounded by the calibration error of Definition 2.3. $\qquad \square$

---

[2]A similar upper-bound on the principal's utility was proved for no-swap-regret agents [20]. While we prove the theorem directly for calibration, an alternative proof in App. B shows that calibration implies no-swap regret.

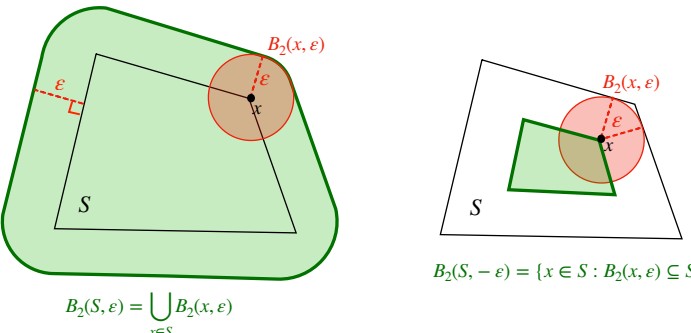

$$B_2(S, \varepsilon) = \bigcup_{x \in S} B_2(x, \varepsilon)$$

$$B_2(S, -\varepsilon) = \{x \in S : B_2(x, \varepsilon) \subseteq S\}$$

Figure 2: Pictorial representation for notation $B_2(S, \varepsilon)$ (left) and $B_2(S, -\varepsilon)$ (right).

On the other hand, it may seem that because the agent's behavior is less specified when she uses calibrated forecasts (as opposed to full knowledge), the principal may only be able to extract much less utility compared to $V^\star$. Again, we show that this is not the case and that there exist algorithms for the principal such that the sequence of strategies $\{\mathbf{h}_t\}_{t \in [T]}$ is asymptotically approaching $V^\star$.

**Theorem 3.2.** *There exists an algorithm for the principal in CSGs that achieves average utility:* $\lim_{T \to \infty} \frac{1}{T} \sum_{t \in [T]} U_P(\mathbf{h}_t, y_t) \geq V^\star$.

Algorithm 1 is an explore-then-commit algorithm; it first estimates an *appropriate* strategy for the principal $\tilde{\mathbf{h}}$ (EXPLORE), and then repeatedly plays it until the end (COMMIT). In the remainder of the section, we sketch the proof for Theorem 3.2 and point to exact lemma statements in the Appendix. Let $T_1, T_2$ denote the set of rounds that belong in the EXPLORE and COMMIT phase respectively.

To elaborate on the objectives of the EXPLORE phase, let us first consider a setting with zero calibration error, where the agent's forecasting algorithm is perfectly and adaptively calibrated, leading to $y_t = \text{BR}(\mathbf{h}_t)$ at every round. The task for the EXPLORE phase simplifies to identifying a near-optimal strategy $\tilde{\mathbf{h}}$ through best response oracles that satisfies $U_P(\tilde{\mathbf{h}}, \text{BR}(\tilde{\mathbf{h}})) \geq V^\star - \varepsilon_1$ for a predetermined $\varepsilon_1$. We formalize this property in **(P1)**. Given that the agent is perfectly calibrated, in the COMMIT phase, the agent always plays $\tilde{y} = \text{BR}(\tilde{\mathbf{h}})$, leading to an upper bound of $\varepsilon_1|T_2|$ on the Stackelberg regret. Hence, Algorithm 1's regret is bounded by $V^\star|T_1| + \varepsilon_1|T_2|$.

**(P1)** $U_P(\tilde{\mathbf{h}}, \tilde{y}) \geq V^\star - \varepsilon_1$ for $\tilde{y} \in \text{BR}(\tilde{\mathbf{h}})$, i.e., $(\tilde{\mathbf{h}}, \tilde{y})$ is an approximate Stackelberg equilibrium.

Moving away from the idealized setting, we must account for possible discrepancies between $y_t$ and $\text{BR}(\mathbf{h}_t)$ due to calibration error. This introduces: (i) An increased sample complexity $|T_1|$ in the EXPLORE phase, given the necessity to learn a near-optimal strategy from noisy responses; (ii) Potential deviations from the action $\tilde{y} = \text{BR}(\tilde{\mathbf{h}})$ in the COMMIT phase due to miscalibrations in belief. To address the first challenge, we employ Algorithm 2, which constructs an approximate best response oracle by repeatedly interacting with a calibrated agent. For the second challenge, we require our learned policy $\tilde{\mathbf{h}}$ to be robust against inaccurate forecasts. This is reflected in condition **(P2)**, which necessitates the ball of radius $\varepsilon_2$ around $\tilde{\mathbf{h}}$ to be fully contained in the polytope $P_{\tilde{y}}$. The critical insight from **(P2)** is: for any forecast $\mathbf{p}_t$ that results in a best response $y_t = \text{BR}(\mathbf{p}_t) \neq \tilde{y}$, there must be a minimum distance of $\varepsilon_2$ separating $\mathbf{p}_t$ from $\tilde{\mathbf{h}}$. We will now proceed to formalize **(P2)**, but before delving into that, it is important to introduce some additional notations.

Let $B_2(x, \varepsilon)$ denote the ball of radius $\varepsilon$ around $x$, i.e., $B_2(x, \varepsilon) \triangleq \{x' : \|x - x'\|_2 \leq \varepsilon\}$. For a convex set $S \in \mathbb{R}^n$, we use $B_2(S, \varepsilon)$ to denote the *union* of all balls of radius $\varepsilon$ around the set, i.e., $B_2(S, \varepsilon) \triangleq \bigcup_{x \in S} B_2(x, \varepsilon)$. For a convex set $S \in \mathbb{R}^n$, we use $B_2(S, -\varepsilon)$ to denote the set of all points in $S$ that are "safely" inside $S$ (i.e., all the points in a ball of radius $\varepsilon$ around them still belong in $S$): $B_2(S, -\varepsilon) \triangleq \{x \in S : B_2(x, \varepsilon) \subseteq S\}$. We call this last set, the $\varepsilon$-*conservative* of $S$. See Figure 2 for a pictorial illustration of the notations.

**Algorithm 1:** Explore-Then-Commit

**Input:** Target precision $\varepsilon$, time horizon $T$.
EXPLORE: Find $\varepsilon$-optimal strategy $\tilde{\mathbf{h}} \in \mathcal{H}_P$ using Algorithm 2.
COMMIT: Repeatedly play $\tilde{\mathbf{h}}$ for the rest of the rounds.

Given the above notations and for tuned $\varepsilon_2$, the pair $(\tilde{\mathbf{h}}, \tilde{y})$ returned by Algorithm 2[3] satisfies property **(P1)** and the additional property **(P2)**:

**(P2)** $\tilde{\mathbf{h}} \in B_2(P_{\tilde{y}}, -\varepsilon_2)$, i.e., $\tilde{\mathbf{h}}$ lies *robustly* within the best-response polytope for $\tilde{y}$.

Given **(P2)**, the regret of the COMMIT phase can be decomposed to when $y_t = \tilde{y}$, and when $y_t \neq \tilde{y}$:

$$\sum_{t \in T_2}(V^\star - U_P(\tilde{\mathbf{h}}, y_t)) = \sum_{t \in T_2 : y_t = \tilde{y}}(V^\star - U_P(\tilde{\mathbf{h}}, y_t)) + \sum_{t \in T_2 : y_t \neq \tilde{y}}(V^\star - U_P(\tilde{\mathbf{h}}, y_t)) \quad (3)$$

When $y_t = \tilde{y}$, **(P1)** guarantees that $V^\star - U_P(\tilde{\mathbf{h}}, \tilde{y}) \leq \varepsilon_1$, so the first term is at most $\varepsilon_1 \cdot |T_2|$.

When $y_t \neq \tilde{y}$, let $A = \mathcal{A}_A \setminus \{\tilde{y}\}$. For $i \in A$, the definition of binning function $w_i$ guarantees that the probability of playing action $i$ on forecast $\mathbf{p}_t$ is exactly $w_i(\mathbf{p}_t)$. Based on this observation, the second term of Equation (3) can be further bounded as

$$\sum_{t \in T_2}\sum_{i \in A} w_i(\mathbf{p}_t)(V^\star - U_P(\tilde{\mathbf{h}}, i)) \leq \sum_{i \in A}\sum_{t \in T_2} w_i(\mathbf{p}_t)V^\star = \sum_{i \in A} n_{T_2}(i)V^\star. \quad (4)$$

Using Definition 2.2 of the calibration error and properties of the $\ell_2$ and $\ell_\infty$ norms, we can further express $n_{T_2}(i)$ as follows

$$n_{T_2}(i) = \frac{\text{CalErr}_i(\mathbf{h}_{T_2}, \mathbf{p}_{T_2}) \cdot |T_2|}{\|\bar{\mathbf{p}}_{T_2}(i) - \tilde{\mathbf{h}}\|_\infty} \leq \frac{\sqrt{m} \cdot \text{CalErr}_i(\mathbf{h}_{T_2}, \mathbf{p}_{T_2}) \cdot |T_2|}{\|\bar{\mathbf{p}}_{T_2}(i) - \tilde{\mathbf{h}}\|_2} \overset{\textbf{(P2)}}{\leq} \frac{\sqrt{m} \cdot r_\delta(|T_2|) \cdot |T_2|}{\varepsilon_2},$$

where for the second inequality is: since $\tilde{\mathbf{h}}$ lies in the $\varepsilon_2$-conservative of $P_{\tilde{y}}$, and $\bar{\mathbf{p}}_{T_2}(i)$ belongs to a different and non-intersecting polytope $P_i$, we know that $\|\bar{\mathbf{p}}_{T_2}(i) - \tilde{\mathbf{h}}\|_2 \geq \varepsilon_2$. See Figure 3 for a geometric interpretation. Finally, bounding the sample complexity for the EXPLORE phase (Lemma C.4) and taking asymptotics gives the result.

The proof sketch above hinges on being able to identify a strategy for the principal $\tilde{\mathbf{h}}$ with properties **(P1)**, **(P2)**. We outline below how this is done through Algorithm 2.

At a high level, Algorithm 2 proceeds as follows. It builds an *initialization set* $\mathcal{I}$ which consists of pairs $(\mathbf{h}, y)$ where $\mathbf{h}$ approximately belongs to the $\varepsilon_2$-*conservative* of $y$'s best response polytope: $B_2(P_y, -\varepsilon_2)$. Using these points in the initialization set $\mathcal{I}$, the algorithm then solves the convex optimization problem of $\max_{\mathbf{h}} U_P(\mathbf{h}, y_i)$ using membership queries to $B_2(P_y, -\varepsilon_2)$ and the points of $\mathcal{I}$ as initial points. The final solution is obtained as the maximum of all these convex programs.

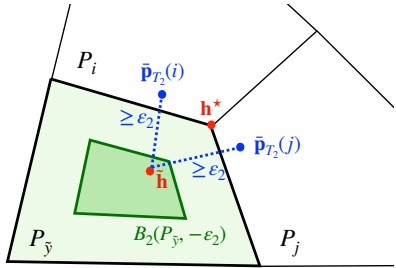

Figure 3: Relationship between the robust policy $\tilde{\mathbf{h}}$ and the average predictions $\bar{\mathbf{p}}_{T_2}(i)$: Given that $\tilde{\mathbf{h}}$ is in the conservative region $B_2(P_{\tilde{y}}, -\varepsilon_2)$, any average prediction $\bar{\mathbf{p}}_{T_2}(i)$ that triggers action $i \neq \tilde{y}$ during the COMMIT phase must fall outside of $P_{\tilde{y}}$ and thus have a distance of at least $\varepsilon_2$ from $\tilde{\mathbf{h}}$.

There are 2 things left to be specified for Algorithm 2; how to build the membership oracle to $B_2(P_y, -\varepsilon_2)$ given that $U_A(\cdot)$ is unknown to the principal, and how to build the initialization set $\mathcal{I}$ to contain at least one point that is well centered in $P_{y^\star}$. To build set $\mathcal{I}$, Algorithm 2 samples strategies $\mathbf{h}_i$ uniformly at random for enough

---

[3]With post-processing if necessary, see Appendix C.5 for more details.

**Algorithm 2:** Principal's Learning Algorithm for the Optimal Commitment

---

Originally, initialization set $\mathcal{I} \leftarrow \emptyset$.

/* build initialization set with pairs $(\mathbf{h}, y)$, where $\mathbf{h} \in B_2(P_y, -\varepsilon_2)$       */

**for** $i \in [O(\log(T)/\textit{Volume}(\eta)]$ **do**

    Sample a strategy $\mathbf{h}_t \in \mathcal{H}_P$ uniformly at random.

    Query APPROXMEM (Algorithm 4) for action $y_i \in \mathcal{A}_A$

    $\mathcal{I} \leftarrow \mathcal{I} \cup (y_i, \mathbf{h}_i)$.

/* optimize $U_P(\mathbf{h}, y)$ using membership queries to $B_2(P_y, -\varepsilon_2)$ and set $\mathcal{I}$     */

**for** $(y_i, \mathbf{h}_i) \in \mathcal{I}$ **do**

    Solve Equation (5) using LSV [42] (initial point: $\mathbf{h}_i$, oracle: $\varepsilon_1$-approximate membership
    oracle to $B_2(P_{y_i}, -\varepsilon_2)$ simulated by APPROXMEM (Algorithm 4)

$$\max_{\mathbf{h}} U_P(\mathbf{h}, y_i), \quad \text{subject to } \mathbf{h} \in B_2(P_{y_i}, -\varepsilon_2), \tag{5}$$

    $\tilde{\mathbf{h}}_{y_i} \leftarrow \varepsilon'$-optimal solution of Equation (5)

$\tilde{y} \leftarrow \arg\max_{y_i \in \mathcal{I}} U_P(\tilde{\mathbf{h}}_{y_i}, y_i)$

RETURN $\tilde{\mathbf{h}}_{\tilde{y}}$

---

number of rounds to ensure at least one sample in the $\eta/2$-ball around the center of $P_{y^\star}$ with high probability. To build the approximate membership oracle for the $\varepsilon_2$-conservative best-response polytope for an action $y$, we use APPROXMEM (Algorithm 4 in Appendix C.2). Specifically, on input $\mathbf{h} \in \mathcal{H}_P$, APPROXMEM either asserts $\mathbf{h} \in B_2(P_y, -\varepsilon_2 + \varepsilon_1)$ or $\mathbf{h} \notin B_2(P_y, -\varepsilon_2 - \varepsilon_1)$ with probability at least $1 - \varepsilon_3$. To do this, it samples $\Phi$ points in proximity to $\mathbf{h}$ and plays each one repeatedly for $l$ rounds, while registering the best-response action observed for each one of these. If the most frequent best-response for all $\mathbf{h}_\phi$ is $y$, then we can conclude with good probability that $\mathbf{h}$ was inside $B_2(P_y, -\varepsilon_2)$. See Lemma C.2 for more details.

# 4   Forecasting Algorithm for Adaptive Calibration

In this section, we examine whether there exist natural forecasting procedures that satisfy our Definition 2.3 about adaptively calibrated forecasts. We answer this question positively.

**Theorem 4.1.** *For all $\varepsilon > 0$ and all binnings $\Pi = \{w_i : \mathbb{R}^m \to [0,1], i \in [k]\}$, there exists a parameter-free forecasting procedure that is $(\varepsilon, \Pi)$-adaptively calibrated with rate $r_\delta(t) = O(\sqrt{\log(kmt)/t})$. Moreover, when $\Pi$ is a continuous binning (i.e., each $w_i$ is continuous), there exists a forecasting procedure that is $(0, \Pi)$-adaptively calibrated with the same rate.*

To prove the theorem, we use two main tools; the first one is a well-known algorithm of Luo and Schapire [44] (ADANORMALHEDGE) applied for online learning in the *sleeping experts* problem (see Appendix D for details). Roughly speaking, the *sleeping experts* is a standard online learning problem with $T$ rounds and $N$ experts, where at each round $t$ there is only a subset of the experts being "awake" to be considered by the learner and report their predictions. Let $I_{t,i}$ be the binary variable indicating whether expert $i$ was awake at round $t$ ($I_{t,i} = 1$) or asleep ($I_{t,i} = 0$). The interaction protocol between the learner and the adversary at each round $t$ is: (i) The learner observes which experts are awake, i.e., $\{I_{t,i}\}_{i \in [N]}$. (ii) The learner selects a probability distribution $\pi_t \in \Delta([N])$ supported on the set of *active* experts $A_t \triangleq \{i : I_{t,i} = 1\}$. (iii) The adversary selects a loss vector $\{\ell_{t,i}\}_{i \in [N]}$. (iv) The learner incurs expected loss $\hat{\ell}_t = \mathbb{E}_{i \sim \pi_t}[\ell_{t,i}]$. ADANORMALHEDGE is a *parameter-free* online learning algorithm that when applied on the sleeping experts problem (and with appropriate initialization) obtains regret $\text{Reg}_t(i) = O(\sqrt{T_i \log(NT_i)})$, where $T_i = \sum_{\tau \in [t]} I_{\tau,i}$.

The second tool that we use is *No-Regret vs. Best-Response dynamics (NRBR)* [34]. NRBR are a form of no-regret dynamics between two players, where one of the players must also best-respond on average. Essentially, at each round $t \in [T]$, the forecasting algorithm with the calibration rate of Theorem 4.1 outputs a randomized forecast $\mathbf{p}_t \in \mathcal{F}_P$, by simulating an interaction between two players described below. For the first player, we construct a *sleeping experts* problem instance, where the set of experts is $\mathcal{G} = \{g_{(s,i,j,\sigma)} : s \in [T], i \in \mathcal{A}_A, j \in \mathcal{A}_P, \sigma \in \{\pm 1\}\}$. For each $g_{(s,i,j,(\sigma))} \in \mathcal{G}$

and $t \in [T]$, we define the loss, sleeping/awake indicator, and instantaneous regret respectively as:

$$\ell_{t,g_{(s,i,j,\sigma)}} \triangleq L_{g_{(s,i,j,\sigma)}}(\mathbf{h}_t, \mathbf{p}_t) = w_i(\mathbf{p}_t) \cdot \sigma \cdot (h_{t,j} - p_{t,j}); \tag{6}$$

$$I_{t,g_{(s,i,j,\sigma)}} \triangleq \mathbf{1}\{t \geq s\}; \tag{7}$$

$$r_{t,g} \triangleq I_{t,g} \cdot (\ell_{t,g} - \hat{\ell}_t).$$

where by $h_{t,j}, p_{t,j}$ we denote the $j$-th coordinate of $\mathbf{h}_t$ and $\mathbf{p}_t$ respectively. We defined the losses for our newly constructed sleeping experts' instance as above to make sure that there is a direct correspondence with the calibration error. Similar ideas for calibration (albeit not for the notion of adaptivity we consider) have been used in [41, 34]. We describe next the player interaction in NRBR.

**Player 1.** Runs ADANORMALHEDGE on expert set $\mathcal{G}$ with a pre-specified prior $\pi_0$ over $\mathcal{G}$ and feedback specified in Equations (6), (7). At each round $t$, Player 1 computes distribution $\pi_t \in \Delta(A_t(\mathcal{G}))$, where $A_t(\mathcal{G})$ denotes the set of active experts $g_{(s,i,j,\sigma)} \in \mathcal{G}$ with $I_{t,g_{(s,i,j,\sigma)}} = 1$.

**Player 2.** Best responds to $\pi_t$ by selecting $Q_t \in \Delta(\mathcal{F}_P)$ that satisfies:

$$\max_{\mathbf{h}_t \in \mathcal{H}_P} \mathbb{E}_{\substack{g \sim \pi_t \\ \mathbf{p}_t \sim Q_t}} [\ell_{t,g}] = \max_{\mathbf{h}_t \in \mathcal{H}_P} \mathbb{E}_{\substack{g \sim \pi_t \\ \mathbf{p}_t \sim Q_t}} [L_g(\mathbf{h}_t, \mathbf{p}_t)] \leq \varepsilon. \tag{8}$$

After simulating the game above, the algorithm outputs forecast $\mathbf{p}_t \sim Q_t$. The existence of such a distribution $Q_t$ is justified by the min-max theorem ([34, Fact 4.1] or [28, Theorem 5]). In the Appendix, we also give an explicit formula for $Q_t$ in the special case of $m = 2$. When $\Pi_0$ is continuous, player 2 can select a deterministic $\mathbf{p}_t$ that achieves Equation (8) with $\varepsilon = 0$. This stronger property is justified by the outgoing fixed-point theorem [28, Theorem 4]. Note that this algorithm inherits its parameter-free property directly from ADANORMALHEDGE. We are now ready to provide a proof sketch for Theorem 4.1.

*Proof sketch of Theorem 4.1.* Fix an instance of the NRBR game outlined above. We begin by Definition 2.2 of calibration error translated in the sleeping experts instance that we defined above:

$$\text{CalErr}_i(\mathbf{h}_{s:t}, \mathbf{p}_{s:t}) \cdot (t - s) = \max_{j \in [m]} \max_{\sigma \in \{\pm 1\}} \sum_{\tau \in [s,t]} I_{\tau, g_{(s,i,j,\sigma)}} \ell_{\tau, g_{(s,i,j,\sigma)}}$$

We add and subtract in the above $\sum_\tau \hat{\ell}_\tau$ to make the regret of the ADANORMALHEDGE on the sleeping experts instance appear and so the aforementioned becomes:

$$\text{CalErr}_i(\mathbf{h}_{s:t}, \mathbf{p}_{s:t}) \cdot (t - s) = \underbrace{\text{Reg}_t(g_{(s,i,j,\sigma)})}_{O(\sqrt{(t-s)\cdot\log(kmt)})} + \max_{j \in \mathcal{A}_P} \max_{\sigma \in \{\pm 1\}} \underbrace{\sum_{\tau \in [s,t]} I_{\tau, g_{(s,i,j,\sigma)}} \hat{\ell}_{\tau, g_{(s,i,j,\sigma)}}}_{\mathcal{T}}$$

where for the regret, we have substituted the regret obtained by ADANORMALHEDGE. Note that $\hat{\ell}_\tau = \mathbb{E}_{g \sim \pi_\tau}[\ell_{\tau,g}]$. This, together with Equation (6) helps us translate the sleeping experts' loss to a loss that depends on $\mathbf{h}_\tau, \mathbf{p}_\tau$. Adding and subtracting the term $\sum_{\tau \in [s,t]} \mathbb{E}_{\substack{g \sim \pi_\tau \\ \mathbf{p} \sim Q_\tau}} [L_g(\mathbf{h}_\tau, \mathbf{p})]$:

$$\mathcal{T} \leq \sum_{\tau \in [s,t]} \max_{\mathbf{h}_\tau \in \Delta(\mathcal{A}_P)} \mathbb{E}_{\substack{g \sim \pi_\tau \\ \mathbf{p} \sim Q_\tau}} [L_g(\mathbf{h}_\tau, \mathbf{p})] + \sum_{\tau \in [s,t]} \mathbb{E}_{g \sim P_\tau} \left[ L_g(\mathbf{h}_\tau, \mathbf{p}_\tau) - \mathbb{E}_{\mathbf{p} \in Q_\tau} [L_g(\mathbf{h}_\tau, \mathbf{p})] \right]$$

The first term above is upper bounded by $\varepsilon \cdot (t - s)$ (Equation (8)) (or 0 in the continuous setting). The second can be bound with martingale concentration inequalities. $\square$

## 5  Continuous Games

In this section, we generalize our results for the case of *continuous* Stackelberg games. The supplementary material can be found in Appendix E.

**Continuous Stackelberg Games.** We use again $\mathcal{A}_P$ and $\mathcal{A}_A$ to denote the principal and the agent action spaces, respectively. Both $\mathcal{A}_A, \mathcal{A}_P$ are convex, compact sets where $\mathcal{A}_P \subset \mathbb{R}^m$ and $\mathcal{A}_A \subset \mathbb{R}^k$. The utilities of the principal and the agent are given by continuous functions $U_P : \mathcal{A}_P \times \mathcal{A}_A \to \mathbb{R}_+$ and $U_A : \mathcal{A}_P \times \mathcal{A}_A \to \mathbb{R}_+$. In this setting, we assume that both the principal and the agent can only play deterministic strategies, i.e., $\mathcal{H}_P = \mathcal{A}_P$. For $x \in \mathcal{A}_P$, let $\text{BR}(x)$ be the best-response function that is implicitly defined as $\nabla_2 U_A(x, \text{BR}(x)) = 0$. Our continuous games satisfy Assumption 5.1: (i)-(iii) are standard assumptions used in previous works (e.g., [24]), but (iv) cannot be derived from (i) and (ii) without further assumptions on the correlation between $x, y$. Nevertheless, (iv) (and the conditions under which it holds) has been justified in settings such as strategic classification [21, 51].

**Algorithm 3:** Lazy Gradient Descent without a Gradient (LAZYGDWOG)

---

Initialize $\mathbf{h}_0 = 0$.

**for** *epoch $\phi \geq 0$:* **do**

    Sample $S_t$ uniformly at random from the unit sphere $\mathbb{S}^{m-1}$.

    Play $\mathbf{h}_\phi = \mathbf{x}_\phi + \delta_\phi S_\phi$ for $M$ rounds.          `/* avg feedback gets close to BR(h_t) */`

    Observe agent's responses $y_{\phi,1}, \cdots, y_{\phi,M}$.

    Update action $\mathbf{x}_{\phi+1} \leftarrow \texttt{Proj}_{B_2(\mathcal{A}_P, -\delta_\phi)}(\mathbf{x}_\phi + \gamma_\phi \frac{m}{\delta_\phi} S_t U_P(\mathbf{h}_\phi, \frac{1}{M} \sum_{i \in [M]} y_{\phi,i}))$.

---

**Assumption 5.1.** *Utility functions $U_P$, $U_A$, and the domain $\mathcal{A}_P$ satisfy the following:*

*(i) For all $x \in \mathcal{A}_P, y \in \mathcal{A}_A$, $U_P(x,y)$ is $L_1$-Lipschitz and concave in $x$, $L_2$-Lipschitz in $y$, and bounded by $W_P$ in $\ell_2$ norm.*

*(ii) The best-response function $\texttt{BR} : \mathcal{A}_P \to \mathcal{A}_A$ is $L_{\texttt{BR}}$-Lipschitz.*

*(iii) Regularity of the feasible set $\mathcal{A}_P = \mathcal{H}_P = \mathcal{F}_P$:*

- *The diameter is bounded: $\texttt{diam}(\mathcal{F}_P) = \sup_{\mathbf{h}, \mathbf{h}' \in \mathcal{F}_P} \|\mathbf{h} - \mathbf{h}'\|_2 \leq D_P$.*

- *$B(0, r) \subseteq \mathcal{A}_P \subseteq B(0, R)$.*

*(iv) The function $U_P(\mathbf{h}, \texttt{BR}(\mathbf{h}))$ is concave with respect to $\mathbf{h}$, and has Lipschitz constant $L_U$.*

The main result of this section is to show that even in *continuous* CSGs, we can approximate asymptotically $V^\star$ for the principal's utility, and that no better utility is actually achieved.

**Theorem 5.2.** *For continuous CSGs satisfying Assumption 5.1, for all $\varepsilon_0 > 0$, there exists a finite binning $\Pi_0$ such that if the agent is $(0, \Pi_0)$[4] - adaptively calibrated and the principal runs an appropriately parametrized instance of LAZYGDWOG (Algorithm 3) then: $\lim_{\substack{\Phi \to \infty \\ M \to \infty}} \frac{1}{\Phi M} \sum_{\phi \in [\Phi]} \sum_{i \in [M]} U_P(\mathbf{h}_\phi, y_{\phi,i}) \geq V^\star - \varepsilon_0$. Moreover, for any sequence of the principal's actions $\mathbf{h}_{[1:T]}$, it holds that: $\lim_{T \to \infty} \frac{1}{T} \sum_{t \in [T]} U_P(\mathbf{h}_t, y_t) \leq V^\star + \varepsilon_0$.*

We outline next how LAZYGDWOG works. LAZYGDWOG is a variant of the gradient descent without a gradient algorithm (GDWOG) of Flaxman et al. [26]. The main new component of the algorithm is that it separates the time horizon into epochs and for each epoch it runs an update of the GDWOG algorithm. During all the rounds that comprise an epoch ($M$ in total), LAZYGDWOG presents the same (appropriately smoothed-out) strategy to the agent and observes the $M$ different responses from the agent. The intuition behind repeating the same strategy for $M$ rounds is that the principal wants to give the opportunity to the agent to recalibrate for a better forecast, i.e., $\lim_{M \to \infty} \frac{1}{M} |\{i \in [M] : \|\mathbf{p}_i - \mathbf{h}\| \geq \varepsilon_0\}| = 0$. The remainder of the proof for Theorem 5.2 focuses on showing that when the calibrated forecasts converge to $\mathbf{h}_t$, then the principal's utility converges to the utility they would have gotten if the agent was perfectly best responding to $\mathbf{h}_t$.

## 6 Discussion and future directions

In this paper we introduced and studied learning in CSGs, where the agents best respond to the principal's actions based on *calibrated* forecasts that they have about them. Our work opens up several exciting avenues for future research. First, although our main results prove asymptotic convergence, it is an open question whether our exact convergence rates can be improved both for general CSGs and for specific cases of Stackelberg games (e.g., strategic classification in more general models compared to [21], pricing [1]). Second, it is an interesting question whether our definition of adaptive calibration (Def. 2.3) can actually hold for the sum over all binning functions, instead of just the maximum. Finally, to provide our asymptotic convergence results we assumed that the principal has access to the agent's calibration rate $r_\delta(\cdot)$; *some* information regarding how $\mathbf{p}_t$'s relate to $\mathbf{h}_t$'s is necessary to leverage the fact that agents are calibrated. But we think that in some specific settings (e.g., strategic classification) there may actually exist extra information regarding the forecasts (compared to just knowing $r_\delta(\cdot)$) that can be leveraged to design learning algorithms for the principal with faster convergence rates. We discuss these directions in more detail in Appendix F.

---

[4]We can define $(0, \Pi)$-adaptive calibration in continuous CSGs due to the continuous case in Theorem 4.1.

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

# A  Calibrated Forecasts Standard Definition

We give below the standard definition for asymptotic calibration of Foster and Vohra [30] for a sequence of binary outcomes, i.e., $\mathbf{h}_t \in A = \{0,1\}, \forall t \in [T]$. The forecasts $\mathbf{p}_t$ take values in $C = [0,1]$. Let $X$ denote the adaptive adversary generating the events' sequence (which is of infinite size), where the $T$ first events are $\mathbf{h}_1, \ldots, \mathbf{h}_T$.

**Definition A.1.** *A forecasting procedure $\sigma$ is* asymptotically calibrated *if and only if for any adaptive adversary $X$ that generates the sequence $\mathbf{h}_1, \cdots, \mathbf{h}_T \in A$ and the forecasting algorithm $\sigma$ that generates (possibly random) forecasts $\mathbf{p}_1, \cdots, \mathbf{p}_T \in C$ on the same sequence, we have that the calibration score $C_T(X, \sigma)$ goes to 0 as $T \to \infty$:*

$$C_T(X, \sigma) \triangleq \sum_{\mathbf{p} \in \mathcal{F}_P} \frac{n_T(\mathbf{p}; \mathbf{h}, \sigma)}{T} \left| \rho_T(\mathbf{p}; \mathbf{h}, \sigma) - \mathbf{p} \right| \tag{9}$$

*where $n_T(\mathbf{p}; \mathbf{h}, \sigma) \triangleq \sum_{t \in [T]} \mathbf{1}\{\mathbf{p}_t = \mathbf{p}\}$ is the number of times that $\sigma$ predicts $\mathbf{p}$ and $\rho_T(\mathbf{p}; \mathbf{h}, \sigma) \triangleq \frac{\sum_{t \in [T]} \mathbf{h}_t \mathbf{1}\{\mathbf{p}_t = \mathbf{p}\}}{n_T(\mathbf{p}; \mathbf{h}, \sigma)}$ be the fraction (empirical probability) of these times that the actual event was 1.*

Note that in Eq. (9), while $\mathcal{F}_P$ contains an infinite number of distinct $\mathbf{p}$'s (hence an infinite number of summands), for every finite $T$, there is only a finite number of $\mathbf{p}$ where $n_T(\mathbf{p}; \mathbf{h}, \sigma)$ is nonzero. Therefore, $C_T$ is well-defined and finite.

Equivalently, the above definition states that for the infinite binning [28] $\Pi = \{w_x(\mathbf{p}) : x \in C\}$ where $w_x(\mathbf{p}) = \mathbf{1}\{\mathbf{p} = x\}$, the calibration score can be equivalently expressed as

$$C_T(X, \sigma) \triangleq \sum_{w_x \in \Pi} \frac{n_T(x)}{T} \left\| \overline{\mathbf{h}}_T(x) - \overline{\mathbf{p}}_T(x) \right\|,$$

where $n_T(x) \triangleq \sum_{t=1}^{T} w_x(\mathbf{p}_t)$ is the number of times that forecast $\mathbf{p}_t$ falls into bin $x$, $\overline{\mathbf{p}}_T(x) \triangleq \sum_{t=1}^{T} \frac{w_x(\mathbf{p}_t)}{n_T(x)} \cdot \mathbf{p}_t$ is the average forecast that activates bin $x$, which is equal to $\sum_{t=1}^{T} \frac{w_x(\mathbf{p}_t)}{n_T(x)} \cdot x = x$ because $w_x(\mathbf{p}_t)$ is nonzero if and only if $\mathbf{p}_t = x$, and $\overline{\mathbf{h}}_T(x) \triangleq \sum_{t=1}^{T} \frac{w_x(\mathbf{p}_t)}{n_T(x)} \cdot \mathbf{h}_t$ is the average outcome corresponding to bin $x$. It follows that the score $C_T$ is a sum of the calibration errors during interval $[1 : T]$ for all bins (with CalErr defined in Definition 2.3).

$$C_T(X, \sigma) = \sum_{w_x \in \Pi} \mathrm{CalErr}_x(\mathbf{h}_{1:T}, \mathbf{p}_{1:T}).$$

# B  Calibrated Forecasts Lead to No Swap Regret

In this section, we show the connection between no-swap-regret agents and adaptively calibrated ones. As a reminder, no-swap-regret agents (translated to our setting and notation for the ease of exposition) are defined as follows.

**Definition B.1** (Agent's swap regret [10]). *For a sequence of principal's strategies $\mathbf{h}_1, \cdots, \mathbf{h}_T \in \mathcal{H}_P$ and agent's actions $y_1, \cdots, y_T \in \mathcal{A}_A$, the swap regret is defined as*

$$\mathrm{SwapReg}(\mathbf{h}_{1:T}, y_{1:T}) = \max_{\pi : \mathcal{A}_A \to \mathcal{A}_A} \sum_{t \in [T]} U_A(\mathbf{h}_t, \pi(y_t)) - \sum_{t \in [T]} U_A(\mathbf{h}_t, y_t).$$

*We say that an agent is a* no-swap-regret *agent, if for the sequence of actions $\{y_t\}_{t \in [T]}$ that they are playing it holds that $\mathrm{SwapReg}(\mathbf{h}_{1:T}, y_{1:T}) = o(T)$.*

We next show that calibrated forecasts lead to no swap regret actions for the agent.

**Lemma B.2** (Calibrated forecasts lead to no swap regret). *If the agent is $(\varepsilon, \Pi)$-adaptively calibrated, then the agent's swap regret on the sequence $\mathbf{h}_{1:T}$ is bounded by the calibration error as follows:*

- *If the agent breaks ties deterministically, then with probability $\geq 1 - \delta$,*

$$\mathrm{SwapReg}(\mathbf{h}_{1:T}, y_{1:T}) \leq 2U_{\max} m k T \left( r_\delta(T) + \varepsilon \right) \in o(T).$$

- *If the agent breaks ties randomly, then with probability $\geq 1 - 2\delta$,*

$$\mathrm{SwapReg}(\mathbf{h}_{1:T}, y_{1:T}) \leq U_{\max} \left( O\left( \sqrt{Tk \log\left(\frac{k}{\delta}\right)} \right) + 2mkT\left(r_\delta(T) + \varepsilon\right) \right) \in o(T).$$

*where $U_{\max} = \max_{\mathbf{h} \in \mathcal{A}_P} \max_{y \in \mathcal{A}_A} U_A(\mathbf{h}, y)$ is the maximum utility the agent can obtain (without constraining the agent to play best responses).*

*Proof.* We first present the proof for the case that the agents break ties deterministically. To simplify notation, we use $n_T(i) := n_{[0:T]}(i)$, $\bar{\mathbf{p}}_T := \bar{\mathbf{p}}_{[0:T]}(i)$, and $\bar{\mathbf{h}}_T(i) := \bar{\mathbf{h}}_{[0:T]}(i)$.

Fix a $\pi : \mathcal{A}_A \to \mathcal{A}_A$. Then, with probability at least $1 - \delta$, we have that:

$$\sum_{t=1}^{T} U_A(\mathbf{h}_t, \pi(y_t)) - \sum_{t=1}^{T} U_A(\mathbf{h}_t, y_t)$$

$$= \sum_{i \in \mathcal{A}_A} \sum_{t=1}^{T} \mathbf{1}\{y_t = i\} \left( U_A(\mathbf{h}_t, \pi(i)) - U_A(\mathbf{h}_t, i) \right) \qquad \text{(rewriting } y_t \text{ as the exact action)}$$

$$\overset{(a)}{=} \sum_{i \in \mathcal{A}_A} \sum_{t=1}^{T} w_i(\mathbf{p}_t) \left( \langle \mathbf{h}_t, U_A(\cdot, \pi(i)) \rangle - \langle \mathbf{h}_t, U_A(\cdot, i) \rangle \right) \qquad (10)$$

$$= \sum_{i \in \mathcal{A}_A} n_T(i) \left( \langle \bar{\mathbf{h}}_T(i), U_A(\cdot, \pi(i)) \rangle - \langle \bar{\mathbf{h}}_T(i), U_A(\cdot, i) \rangle \right)$$

$$= \sum_{i \in \mathcal{A}_A} n_T(i) \left( \langle \bar{\mathbf{h}}_T(i) - \bar{\mathbf{p}}_T(i), U_A(\cdot, \pi(i)) \rangle + \langle \bar{\mathbf{p}}_T(i), U_A(\cdot, \pi(i)) - U_A(\cdot, i) \rangle + \langle \bar{\mathbf{p}}_T(i) - \bar{\mathbf{h}}_T(i), U_A(\cdot, i) \rangle \right)$$

$$\overset{(b)}{\leq} \sum_{i \in \mathcal{A}_A} n_T(i) \left\| \bar{\mathbf{p}}_T(i) - \bar{\mathbf{h}}_T(i) \right\|_\infty \left( \| U_A(\cdot, \pi(i)) \|_1 + \| U_A(\cdot, i) \|_1 \right)$$

$$= 2U_{\max} m \cdot \sum_{i \in \mathcal{A}_A} T \cdot \mathrm{CalErr}_i(\mathbf{h}_{1:T}, \mathbf{p}_{1:T}) \qquad \text{(Def. 2.3)}$$

$$\leq 2U_{\max} mkT\left(r_\delta(T) + \varepsilon\right).$$

In the above equations, step (a) is due to the fact that agents best respond with a deterministic tie-breaking rule: $y_t = i$ if and only if $i \in \mathrm{BR}(\mathbf{p}_t)$ and $i \succ j, \forall j \neq i$, which is equivalent to $w_i(\mathbf{p}_t) = 1$. We have also used $U_A(\cdot, i)$ to denote the $m$-dimensional vector where the $j$th entry is the utility $U_A(j, i)$. Step (b) is because the second term

$$\langle \bar{\mathbf{p}}_T(i), U_A(\cdot, \pi(i)) - U_A(\cdot, i) \rangle = U_A(\bar{\mathbf{p}}_T(i), \pi(i)) - U_A(\bar{\mathbf{p}}_T(i), i)$$

is non-positive since each $\mathbf{p}_t$ with $w_i(\mathbf{p}_t) = 1$ belongs to the best response polytope $P_i$, so does their average: $\bar{\mathbf{p}}_t(i) \in P_i \iff i \in \mathrm{BR}(\bar{\mathbf{p}}_t(i))$.

Since the above inequality holds for any $\pi$, it also holds after taking the maximum over all $\pi : \mathcal{A}_A \to \mathcal{A}_A$. Therefore, we have the same bound for the agent's swap regret.

Next, we move to the case when the agent breaks ties randomly. For a fixed $\pi$, we have that at every time step $t$,

$$\mathbb{E}_{t-1}\left[ U_A(\mathbf{h}_t, \pi(y_t)) \right] = \frac{\sum_{i \in \mathrm{BR}(\mathbf{p}_t)} U_A(\mathbf{h}_t, \pi(i))}{|\mathrm{BR}(\mathbf{p}_t)|} = \sum_{i \in \mathcal{A}_A} w_i(\mathbf{p}_t) U_A(\mathbf{h}_t, \pi(i)).$$

Therefore, by Azuma-Hoeffding's inequality, w.p. $\geq 1 - \delta'$, we have

$$\sum_{t=1}^{T} U_A(\mathbf{h}_t, \pi(y_t)) \leq \sum_{t=1}^{T} w_i(\mathbf{p}_t) \sum_{i \in \mathcal{A}_A} U_A(\mathbf{h}_t, \pi(i)) + O\left( \sqrt{T \log\left(\frac{1}{\delta'}\right)} \right). \qquad (11)$$

Since all actions in $\mathrm{BR}(\mathbf{p}_t)$ have the same utility for the agents, we also have

$$U_A(\mathbf{h}_t, y_t) = \frac{\sum_{i \in \mathrm{BR}(\mathbf{p}_t)} U_A(\mathbf{h}_t, i)}{|\mathrm{BR}(\mathbf{p}_t)|} = \sum_{i \in \mathcal{A}_A} w_i(\mathbf{p}_t) U_A(\mathbf{h}_t, i).$$

Therefore, using Equations (11) and (12), we have that with probability at least $1 - \delta'$,

$$\sum_{t=1}^{T} U_A(\mathbf{h}_t, \pi(y_t)) - \sum_{t=1}^{T} U_A(\mathbf{h}_t, y_t) \leq \sum_{i \in \mathcal{A}_A} \sum_{t=1}^{T} w_i(\mathbf{p}_t) \left( U_A(\mathbf{h}_t, \pi(i)) - U_A(\mathbf{h}_t, i) \right)$$

$$+ O\left( \sqrt{T \log\left(\frac{1}{\delta'}\right)} \right). \tag{12}$$

We can use the same arguments as above (from Equation (10) onwards) to bound the first term on the right hand side by $2U_{\max} mkT (r_\delta(T) + \varepsilon)$ with probability $1 - \delta$. Finally, setting $\delta' = \delta/M$ where $M = k^k$ is the number of possible swap functions, and applying the union bound, we conclude that with probability $\geq 1 - 2\delta$, the swap regret is bounded by

$$\mathrm{SwapReg}(\mathbf{h}_{1:T}, y_{1:T}) \leq U_{\max} \left( O\left( \sqrt{T \left( k \log k + \log\left(\frac{1}{\delta}\right) \right)} \right) + 2mkT (r_\delta(T) + \varepsilon) \right).$$

$\square$

# C    Supplementary Material for Section 3

## C.1    Proof of Theorem 3.1

We first formally restate the theorem with exact convergence rates (not just asymptotically).

**Theorem C.1** (Formal version of Theorem 3.1)**.** *Assume that the agent is $(\varepsilon, \Pi)$-adaptively calibrated with rate $r_\delta(\cdot)$ and let $U_{\max} = \max_{\mathbf{h} \in \mathcal{H}_P} \max_{y \in \mathcal{A}_A} U_P(\mathbf{h}, y)$. Then, for any sequence $\{\mathbf{h}_t\}_{t \in [T]}$ for the principal's strategies in a repeated CSG, with probability at least $1 - 2\delta$, the principal's utility is upper bounded as:*

$$\sum_{t \in [T]} U_P(\mathbf{h}_t, y_t) \leq V^\star T + \alpha(U_{\max}, m, k, T, r_\delta, \delta, \varepsilon)$$

*where $\alpha(U_{\max}, m, k, T, r_\delta, \delta, \varepsilon) = U_{\max} \cdot m \cdot k \cdot T \cdot (r_\delta(T) + \varepsilon)$ when the agent uses* deterministic *tie-breaking and $\alpha(U_{\max}, m, k, T, r_\delta, \delta, \varepsilon) = U_{\max} \cdot m \cdot k \cdot T \cdot (r_\delta(T) + \varepsilon + \sqrt{T \log(1/\delta)})$ when the agent uses* randomized *tie-breaking.*

*Proof.* To simplify notation, we use $n_T(i) := n_{[0,T]}(i)$. When the agent follows deterministic tie-breaking, we have:

$$\sum_{t=1}^{T} U_P(\mathbf{h}_t, y_t) = \sum_{i \in \mathcal{A}_A} \sum_{t=1}^{T} w_i(\mathbf{p}_t) U_P(\mathbf{h}_t, i)$$

$$= \sum_{i \in \mathcal{A}_A} n_T(i) U_P(\bar{\mathbf{h}}_T(i), i) \qquad \text{(linearity of } U_P \text{ in the principal's strategy)}$$

$$= \sum_{i \in \mathcal{A}_A} n_T(i) \left( U_P\left(\bar{\mathbf{p}}_T(i), i\right) + \left\langle U_P(\cdot, i), \bar{\mathbf{h}}_T(i) - \bar{\mathbf{p}}_T(i) \right\rangle \right)$$

$$(\pm \sum_i U_P(\bar{\mathbf{p}}_T(i), i))$$

$$\overset{(a)}{\leq} \sum_{i \in \mathcal{A}_A} n_T(i) V^\star + \sum_{i \in \mathcal{A}_A} n_T(i) \left\| \bar{\mathbf{p}}_T(i) - \bar{\mathbf{h}}_T(i) \right\|_\infty \cdot \left\| U_P(\cdot, i) \right\|_1$$

$$\overset{(b)}{\leq} V^\star T + U_{\max} m \sum_{i \in \mathcal{A}_A} T \cdot \mathrm{CalErr}_i(\mathbf{h}_{1:T}, \mathbf{p}_{1:T})$$

$$\leq V^\star T + U_{\max} \cdot k \cdot m \cdot T \cdot (r_\delta(T) + \epsilon)$$

where (a) is because $U_P(\bar{\mathbf{p}}_T(i), i) \leq V^\star$ (since $i \in \mathrm{BR}(\bar{\mathbf{p}}_T)$), the Hö(lder's inequality, and the fact that $\| \cdot \|_2 \leq \| \cdot \|_1$, and (b) is because of the definition of $U_{\max}$ and Definition 2.3. The proof for the randomized tie-breaking setting has an extra term from Azuma-Hoeffding's inequality, similar to the proof of Theorem B.2. $\square$

## C.2 Approximate membership oracle to conservative polytopes

In this section, we formally present the algorithm (Algorithm 4 for constructing an approximate membership oracle to the conservative best response polytope for each of the agent's action. The sample complexity of the oracle will be presented in Lemma C.2.

---

**Algorithm 4:** Approximate membership oracle for the conservative best response polytope (APPROXMEM)

---

**Input:** query $\mathbf{h} \in \mathcal{H}_P$, original polytope $P_y$ ($y \in \mathcal{A}_A$), approximation factor $\varepsilon_1$, conservatism factor $\varepsilon_2$, failure probability $\varepsilon_3$

**Parameters:** Number of epochs $\Phi$, radius $R$, calibration error $\varepsilon_{\text{cal}}$.

Let $l$ be s.t. $r_\delta(l) = \frac{\varepsilon_{\text{cal}}}{k\sqrt{m}}$

**for** *epoch* $\phi \in [\Phi]$ **do**

    Sample a point $\mathbf{h}_\phi$ such that $\|\mathbf{h}_\phi - \mathbf{h}\|_2 = R$.        `/* Sample point `$\mathbf{h}_\phi$` close to h */`

    **if** $\mathbf{h}_\phi \notin \mathcal{H}_P$ **then**

        RETURN FALSE        `/* No longer inside the feasible set */`

    **else**

        Play strategy $\mathbf{h}_\phi$ for $l$ rounds.

        $y_\phi \leftarrow$ most frequent best-response action from agent during the $l$ rounds.

        **if** $y_\phi \neq y$ **then** RETURN FALSE        `/* `$\mathbf{h}_\phi$` is too close to `$P_{y_\phi}$` */`

`/* For membership, output `$y_\Phi$` if all `$\{y_\phi\}_{\phi \in [\Phi]}$` agree, and N/A otherwise        */`

RETURN TRUE

---

In Lemma C.2, we show that the parameters $\Phi, \varepsilon_{\text{cal}}, R$ can be tuned to achieve a wide range of parameters $(\varepsilon_1, \varepsilon_2, \varepsilon_3)$. In Proposition C.3, we provide explicit instantiations of the parameters for three special cases of $(\varepsilon_1, \varepsilon_2, \varepsilon_3)$ that will be useful in later sections.

**Lemma C.2.** *If the agent is $(\varepsilon, \Pi)$-adaptively calibrated with rate $r_\delta(\cdot)$ and infinitesimal $\varepsilon$, and the parameters $\Phi, R, \varepsilon_{cal}$ satisfy:*

$$\varepsilon_1 + \varepsilon_2 - R \geq \varepsilon_{cal} \tag{Condition 1}$$

$$\Phi \geq 10\sqrt{m} \left( 1 - \left( \frac{\varepsilon_{cal} + \varepsilon_2}{R} \right)^2 \right)^{\frac{m-1}{2}} \log\left( \frac{1}{\varepsilon_3 - \delta} \right) \tag{Condition 2}$$

*Then Algorithm 4 (APPROXMEM) returns an $\varepsilon_1$-approximate membership oracle to $P_y^{-\varepsilon_2} = B_2(P_y, -\varepsilon_2)$ with probability $1 - \varepsilon_3$, using no more than $N_{\varepsilon_1, \varepsilon_2, \varepsilon_3} = O\left( \Phi r_\delta^{-1}(\frac{\varepsilon_{cal}}{k\sqrt{m}}) \right)$ rounds of interactions with the agent.*

*Specifically, with probability $1 - \varepsilon_3$, APPROXMEM either returns TRUE which asserts that $\mathbf{h} \in B_2(P_y^{-\varepsilon_2}, +\varepsilon_1)$, or returns FALSE which asserts that $\mathbf{h} \notin B_2(P_y^{-\varepsilon_2}, -\varepsilon_1)$.*

**Proposition C.3** (Parameter settings). *Assume $\varepsilon_3 = \delta + T^{-2}$. In the following three cases: $\frac{\varepsilon_1}{\varepsilon_2} = \Theta(\sqrt{m}), 1, o(1)$, the proposed setting of parameters $\varepsilon_{cal}, R, \Phi$ satisfy (Condition 1) and (Condition 2) simultaneously.*

*Case I. $\frac{\varepsilon_1}{\varepsilon_2} = \Theta(\sqrt{m})$.*

$$\varepsilon_{cal} = \varepsilon_2, \quad R = \frac{\varepsilon_1}{2}, \quad \Phi = 10\sqrt{m} \left( 1 - \frac{4\varepsilon_2}{\varepsilon_1} \right)^{\frac{m-1}{2}} \log(T) = O(\sqrt{m}\log(T)).$$

*Case II. $\frac{\varepsilon_1}{\varepsilon_2} = 1$.*

$$\varepsilon_{cal} = 0.1\varepsilon_2, \quad R = 1.9\varepsilon_2, \quad \Phi = 1.25^m \log(T).$$

*Case III. $\frac{\varepsilon_1}{\varepsilon_2} = o(1)$.*

$$\varepsilon_{cal} = \frac{\varepsilon_1}{6}, \quad R = \left( \varepsilon_2 + \frac{\varepsilon_1}{6} \right) \left( 1 + \frac{\varepsilon_1}{2\varepsilon_2} \right), \quad \Phi = 10\sqrt{m} \left( \frac{\varepsilon_2}{\varepsilon_1} \right)^{\frac{m}{2}} \log(T).$$

*Proof of Lemma C.2.* Before we delve into the specifics of the proof, we introduce some notation. In order to make sure that $\mathbf{h}_\phi$ is such that $\|\mathbf{h} - \mathbf{h}_\phi\| = R$, we do the following: $\mathbf{h}_\phi \leftarrow \mathbf{h} + R\mathbf{S}_\phi$, where $\mathbf{S}_\phi$ is sampled uniformly at random from the equator $\mathbb{S} \cap \mathbb{H}$, where $\mathbb{S} = \{\mathbf{s} \in \mathbb{R}^m : \|\mathbf{s}\|_2^2 = 1\}$ is the unit sphere and $\mathbb{H} = \{\mathbf{s} \in \mathbb{R}^m : \langle \mathbf{s}, \mathbf{1} \rangle = 0\}$ is an equatorial hyperplane ($\mathbf{1} \triangleq (1, \cdots, 1) \in \mathbb{R}^m$). Note that this is because we want $\mathbf{h}_\phi$ to remain a valid probability distribution, i.e., that $\langle \mathbf{h}_\phi, \mathbf{1} \rangle = 1$ and $\mathbf{h}_\phi \geq 0$ coordinate-wise; indeed, since we already have $\langle \mathbf{h}, \mathbf{1} \rangle = 1$, we need to make sure that (1) $\langle \mathbf{S}_\phi, \mathbf{1} \rangle = 0$, which is guaranteed by $\mathbf{S}_\phi \in \mathbb{H}$; (2) $\mathbf{h}_\phi \geq 0$, which is guaranteed by returning FALSE whenever $\mathbf{h}_\phi \notin \mathcal{H}_P$.

For the rest of the proof, we condition on the following success event:

$$\mathcal{E} \triangleq \Big\{ \forall [s,t] \subseteq [1,T], \ \forall i \in \mathcal{A}_A, \ \mathrm{CalErr}_i\left(\mathbf{h}_{s:t}, \mathbf{p}_{s:t}\right) \leq r_\delta(t-s) \Big\}.$$

Since the agent is $(\varepsilon, \Pi)$-adaptively calibrated with rate $r_\delta$, we have $\Pr(\mathcal{E}) \geq 1 - \delta$.

Recall that $\varepsilon_{\mathrm{cal}} = k\sqrt{m}r_\delta(l)$ denote the error that comes from calibration error. We first show that conditioned on $\mathcal{E}$, $\forall \phi \in [\Phi]$, $\mathbf{h}_\phi \in B_2(P_{y_\phi}, \varepsilon_{\mathrm{cal}})$. Let $l_{y_\phi}$ be the number of times that agent plays $y_\phi$ during the $l$ repeats, then we have $l_{y_\phi} \geq l/k$ because $y_\phi$ is the most frequently played action. Then, the calibration error bound in Definition 2.3 guarantees that

$$\frac{l_{y_\phi}}{l}\|\bar{\mathbf{p}}(y_\phi) - \mathbf{h}_\phi\|_\infty = \mathrm{CalErr}_{y_\phi}(\mathbf{h}_{\phi,1:l}, \mathbf{p}_{\phi,1:l}) \leq r_\delta(l) + \varepsilon$$

$$\Rightarrow \|\bar{\mathbf{p}}(y_\phi) - \mathbf{h}_\phi\|_2 \leq \sqrt{m}\,\|\bar{\mathbf{p}}(y_\phi) - \mathbf{h}_\phi\|_\infty \leq \sqrt{m}kr_\delta(l) = \varepsilon_{\mathrm{cal}}. \tag{13}$$

where the first inequality in equation (13) is because of the norm property $\|x\|_2 \leq \sqrt{d}\|x\|_\infty$ for a vector $x \in \mathbb{R}^d$. Since $\bar{\mathbf{p}}(y_\phi) \in P_{y_\phi}$ because the agent always best responds to forecasts, we obtain $\mathbf{h}_\phi \in B_2(P_{y_\phi}, \varepsilon_{\mathrm{cal}})$.

We then prove the following two claims:

**(C1)** If $\mathbf{h} \in B_2(P_y^{-\varepsilon_2}, -\varepsilon_1)$, then APPROXMEM returns TRUE.

**(C2)** If $\mathbf{h} \notin B_2(P_y^{-\varepsilon_2}, +\varepsilon_1)$, then APPROXMEM returns FALSE with probability $\geq 1 - (\varepsilon_3 - \delta)$.

Indeed, if the following two claims hold, then we have established that APPROXMEM asserts one of two cases correctly with probability $\geq 1 - (\varepsilon_3 - \delta)$ conditioned on $\mathcal{E}$. Together with the fact that $\Pr(\mathcal{E}) \geq 1 - \delta$, proves the lemma.

**Proof of (C1).** Suppose $\mathbf{h} \in B_2(P_y^{-\varepsilon_2}, -\varepsilon_1)$. Then we know that $B_2(\mathbf{h}, \varepsilon_1) \subseteq P_y^{-\varepsilon_2} = B_2(P_y, -\varepsilon_2)$, which further implies $B_2(\mathbf{h}, \varepsilon_1 + \varepsilon_2) \subseteq P_y$. Therefore, the distance between $\mathbf{h}$ and any other polytope $P_{y'}$ ($y' \neq y$) must be lower bounded by $\varepsilon_1 + \varepsilon_2$. By triangle inequality, $\forall \mathbf{h}' \in P_{y'}$ where $y' \neq y$, we have

$$\forall \phi \in [\Phi], \quad \|\mathbf{h}_\phi - \mathbf{h}'\|_2 \geq \|\mathbf{h} - \mathbf{h}'\|_2 - \|\mathbf{h} - \mathbf{h}_\phi\|_2 \geq \varepsilon_1 + \varepsilon_2 - R \geq \varepsilon_{\mathrm{cal}},$$

where the last step follows from (Condition 1).

Since this holds for all $\mathbf{h}' \in P_{y'}$, it implies $\mathbf{h}_\phi \notin B_2(P_{y'}, \varepsilon_{\mathrm{cal}})$ whenever $y' \neq y$. Together with the fact that $\mathbf{h}_\phi \in B_2(P_{y_\phi}, \varepsilon_{\mathrm{cal}})$, we must have $y_\phi = y$ for all epochs $\phi \in [\Phi]$. Therefore, APPROXMEM always returns TRUE.

**Proof of (C2).** Suppose $\mathbf{h} \notin B_2(P_y^{-\varepsilon_2}, +\varepsilon_1)$. We first analyze the probability of returning FALSE for a fixed epoch $\phi \in [\Phi]$. Let $\partial S$ be the boundary of a convex set $S$, then by triangle inequality, we have

$$d(\mathbf{h}, \partial B_2(P_y, \varepsilon_{\mathrm{cal}})) \leq d(\mathbf{h}, \partial P_y) + d(\partial P_y, \partial B_2(P_y, \varepsilon_{\mathrm{cal}})).$$

For the first term, we have $d(\mathbf{h}, \partial P_y) \leq \varepsilon_2$ since

$$\mathbf{h} \notin B_2(P_y^{-\varepsilon_2}, +\varepsilon_1) \Rightarrow h \notin P_y^{-\varepsilon_2} = B_2(P_y, -\varepsilon_2) \Rightarrow d(\mathbf{h}, P_y) \leq \varepsilon_2.$$

For the second term, we have

$$d(\partial P_y, \partial B_2(P_y, \varepsilon_{\mathrm{cal}})) \leq \varepsilon_{\mathrm{cal}},$$

because $P_y$ is a convex set. Combining the two bounds, we know that $d(\mathbf{h}, \partial B_2(P_y, \varepsilon_{\mathrm{cal}})) \leq \varepsilon_{\mathrm{cal}} + \varepsilon_2$.

Since $\mathbf{h}_\phi$ is uniformly sampled from the sphere of radius $R$ around $\mathbf{h}$, by convexity of $B_2(P_y, \varepsilon_{\mathrm{cal}})$ and the rotation invariance property of a unit sphere, we have

$$\Pr[\mathbf{h}_\phi \notin B_2(P_y, \varepsilon_{\mathrm{cal}})] \geq \Pr[\langle R\mathbf{S}_\phi, \mathbf{v} \rangle \geq \varepsilon_{\mathrm{cal}} + \varepsilon_2] = \Pr\left[\langle \mathbf{S}_\phi, \mathbf{e}_1 \rangle \geq \varepsilon_{\mathrm{cal}+\varepsilon_2}/R\right],$$

where $\mathbf{v}$ is a unit vector pointing in the direction of (one of) the projection from $\mathbf{h}$ to $\partial B_2(P_y, \varepsilon_{\text{cal}})$, and $\mathbf{e}_1 = (1, 0, \cdots, 0) \in \mathbb{R}^m$. According to [23, Lemma 9], we can further lower bound the probability by

$$\Pr[\mathbf{h}_\phi \notin B_2(P_y, \varepsilon_{\text{cal}})] \geq \frac{1}{2\sqrt{m}} \left( 1 - \left( \frac{\varepsilon_{\text{cal}} + \varepsilon_2}{R} \right)^2 \right)^{\frac{m-1}{2}}.$$

Finally, the probability that no epoch returns FALSE (failure of APPROXMEM) is at most

$$(1 - \delta')^\Phi \leq \varepsilon_3 - \delta, \text{ where } \delta' = \frac{1}{2\sqrt{m}} \left( 1 - \left( \frac{\varepsilon_{\text{cal}} + \varepsilon_2}{R} \right)^2 \right)^{\frac{m-1}{2}}.$$

Since $\delta' \ll 1$, the above inequality holds as long as

$$\Phi \geq \frac{2}{\delta'} \log \left( \frac{1}{\varepsilon_3 - \delta} \right) = 4\sqrt{m} \left( 1 - \left( \frac{\varepsilon_{\text{cal}} + \varepsilon_2}{R} \right)^2 \right)^{\frac{m-1}{2}} \log \left( \frac{1}{\varepsilon_3 - \delta} \right),$$

which follows from (Condition 2). Therefore, $\mathbf{h} \notin B_2(P_y^{-\varepsilon_2}, +\varepsilon_1)$, then APPROXMEM returns FALSE with probability $\geq 1 - (\varepsilon_3 - \delta)$.

$\square$

## C.3 Complexity of the explore algorithm

**Lemma C.4** (Sample complexity of Algorithm 2). *Using Algorithm 2, the principal can find with probability at least $1 - (\delta + \varepsilon_{opt} + T^{-1})$, a strategy $\tilde{\mathbf{h}} \in \mathcal{H}_P$ that satisfies the two properties in Lemma C.5. We refer the reader to Theorems C.8 and C.13 for the sample complexity under two different instantiations of Algorithm 2.*

**Lemma C.5** (Optimality and robustness of output policy). *Under the conditions of Lemma C.4, the output $\tilde{\mathbf{h}}$ of Algorithm 2 satisfies the following two properties:*

**(P1)** $U_P(\tilde{\mathbf{h}}, y^\star) \geq V^\star - \varepsilon_{opt}$, *i.e., $\tilde{\mathbf{h}}$ is an approximate Stackelberg equilibrium.*

**(P2)** $\tilde{\mathbf{h}} \in B_2(P_{y^\star}, -\varepsilon_{robust})$ , *i.e., $\tilde{\mathbf{h}}$ lies robustly within the best-response polytope for $y^\star$.*

*We refer the reader to Theorems C.8 and C.13 for the exact values of $\varepsilon_{robust}, \varepsilon_{opt}$ under two different instantiations of Algorithm 2.*

Before we formally state the proof of Lemmas C.4 and C.5, we first show the sample complexity guarantee of the exploration phase of Algorithm 2 (Lemma C.6), then state a useful lemma from prior work on convex optimization from membership queries [42] (Lemma C.7).

**Lemma C.6** (Initialization set of Algorithm 2). *Let $V$ be the volume of an $\ell_2$ ball of radius $\frac{\eta}{2}$ in $\mathbb{R}^m$. Suppose the principal samples $\mathbf{h}$ uniformly from $\mathcal{H}_P$ for $O(V^{-1} \log T)$ times and calls APPROXMEM($\varepsilon_1 = \varepsilon_2 = \eta/4$, $\varepsilon_3 = \delta + T^{-2}/2$) for the membership of each of them, then with probability at least $1 - \delta - T^{-1}$, the initialization set $\mathcal{I}$ contains $(\mathbf{h}_0, y_0)$ where $y_0 = y^\star$ is the optimal target, and $\mathbf{h}_0$ is $\frac{\eta}{2}$-centered in $P_{y^\star}$, i.e., $\mathbf{h}_0 \in B_2(P_{y^\star}, -\frac{\eta}{2})$.*

*The total number of samples required for the initialization phase is $O\left( V^{-1} 1.25^m r_\delta^{-1}(\frac{\eta}{k\sqrt{m}}) \log^2 T \right)$.*

*Proof of Lemma C.6.* By regularity assumption, there exists $\dot{\mathbf{h}} \in P_{y^\star}$, s.t. $B_2(\dot{\mathbf{h}}, \eta) \in P_{y^\star}$. Therefore, $\forall \mathbf{h}' \in B_2(\dot{\mathbf{h}}, \frac{\eta}{2})$, we have $\dot{\mathbf{h}} \in B_2(P_{y^\star}, -\frac{\eta}{2})$. Moreover, since $\mathbf{h}'$ lies robustly inside $P_{y^\star}$, on the success event of APPROXMEM($\varepsilon_1 = \varepsilon_2 = \frac{\eta}{4}$), APPROXMEM($\mathbf{h}'$) must return membership $y^\star$, otherwise we must have $\mathbf{h}' \notin B_2(P_{y^\star}^{-\frac{\eta}{4}}, -\frac{\eta}{4}) = B_2(P_{y^\star}, -\frac{\eta}{2})$, a contradiction. Since the set of all such $\mathbf{h}'$ takes up nontrivial volume $V$ in $\mathcal{H}_P$, we know that $O(V^{-1} \log T)$ uniform samples are guaranteed to hit one with probability $1 - \frac{1}{(2T)}$.

Now we upper bound the failure probability of the initialization phase. Since the agent is miscalibrated with probability $\delta$, random sampling fails to discover centered $\mathbf{h}'$ with probability $\frac{1}{(2T)}$, and the

probability that one of the answers from APPROXMEM is wrong with probability $\frac{1}{(2T)}$, the total failure probability is $1 - \delta - T^{-1}$ as desired.

As for the sample complexity, note that according to Lemma C.2 and proposition C.3, each call to APPROXMEM takes $O\left(1.25^m r_\delta^{-1}(\frac{\eta}{k\sqrt{m}}) \log T\right)$ samples, and the initialization phase calls APPROXMEM for $O(V^{-1} \log T)$ times, the total sample complexity is $O\left(V^{-1} 1.25^m r_\delta^{-1}(\frac{\eta}{k\sqrt{m}}) \log^2 T\right)$. □

**Lemma C.7** (LSV performance guarantee, Theorems 14 and 15 in [42])**. *Let $K$ be a convex set specified by a membership oracle, a point $x_0 \in \mathbb{R}^n$, and $\eta > 0$ such that $B_2(x_0, \eta/2) \subseteq K \subseteq B_2(x_0, 1)$. There exists a universal constant $\gamma_0 > 1$ such that for any convex function $f$ given by an evaluation oracle and any $\varepsilon' > 0$, there is a randomized algorithm that computes a point $z \in B_2(K, \varepsilon')$ such that with probability $\geq 1 - \varepsilon'$,*

$$f(z) \leq \min_{x \in K} f(x) + \varepsilon' \cdot \left(\max_{x \in K} f(x) - \min_{x \in K} f(x)\right),$$

*with constant probability using $O\left(n^2 \log^{O(1)}(n/\varepsilon'\eta)\right)$ calls to the $\varepsilon_1$-approximate membership oracle and evaluation oracle, where $\varepsilon_1 = (\varepsilon'\eta/n)^{\gamma_0}$.*

Note that in Lemma C.7, the required accuracy ($\varepsilon_1$) of the approximate membership oracle is orders of magnitudes smaller than the suboptimality ($\varepsilon$) of the output solution, i.e., $\varepsilon_1 \ll \varepsilon$. Let $\tilde{\mathbf{h}}$ denote the $\varepsilon$-optimal solution returned by LSV. Since we use APPROXMEM which is an $\varepsilon_1$-approximate membership oracle to $K = P_y^{-\varepsilon_2}$, $\tilde{\mathbf{h}}$ will lie in $B_2(K, \varepsilon) = B_2(P_y^{-\varepsilon_2}, \varepsilon) = B_2(P_y, -\varepsilon_2 + \varepsilon)$. In order to find a near-optimal solution that lies robustly within $P_y$, we can either simulate APPROXMEM with more epochs to guarantee $\varepsilon_1 \ll \varepsilon \approx \varepsilon_2$ (see Appendix C.4 for more details), or perform a post-process to $\tilde{\mathbf{h}}$ that pushes it further inside the polytope (see Appendix C.5 for more details).

## C.4 More epochs to guarantee robustness

In this section, we show how to use more epochs in the simulation of each APPROXMEM oracle to obtain higher accuracy feedbacks. This approach directly guarantees that the output of LSV lies robustly within the polytope.

**Theorem C.8.** *If the principal uses Algorithm 2 with parameters APPROXMEM($\varepsilon_1 = (\frac{\varepsilon'\eta}{n})^{\gamma_0}, \varepsilon_2 = 2\varepsilon', \varepsilon_3 = \delta + T^{-2}$) and LSV($\varepsilon'$), then with probability at least $1 - \delta - T^{-1} - \varepsilon'$, the final solution $\tilde{\mathbf{h}}$ satisfies the following two properties:*

**(P1)** $U_P(\tilde{\mathbf{h}}, y^\star) \geq V^\star - 3\varepsilon'$, *i.e., $\tilde{\mathbf{h}}$ is an approximate Stackelberg equilibrium.*

**(P2)** $\tilde{\mathbf{h}} \in B_2(P_{y^\star}, -\varepsilon')$, *i.e., $\tilde{\mathbf{h}}$ lies robustly within the best-response polytope for $y^\star$.*

*The total number of samples needed is $\tilde{O}\left(V^{-1} m^{\frac{5 + m\gamma_0}{2}} (\eta\varepsilon')^{-\frac{m\gamma_0}{2}} r_\delta^{-1}(\frac{(\varepsilon'\eta)^{\gamma_0}}{km^{\gamma_0}\sqrt{m}})\right).$*

*Proof.* We first analyze the failure probability of the algorithm. Since the agent's adaptive calibration error is uniformly bounded by $r_\delta(\cdot)$ with probability $1 - \delta$, the probability that there exists an incorrect answer from APPROXMEM is bounded by $T \cdot T^{-2} = T^{-1}$ conditioned on agent's calibration error being bounded, and the probability that LSV returns a bad solution is bounded by $\varepsilon'$, the total failure probability of the described algorithm is at most $\delta + T^{-1} + \varepsilon'$.

Now we consider the optimization problem for the optimal polytope $P_{y^\star}$, with initial point $\mathbf{h}_0$ that is $\frac{\eta}{2}$-centered in $P_{y^\star}$. There exists such a pair $(\mathbf{h}_0, y^\star) \in \mathcal{I}$ according to Lemma C.6. Let $\tilde{\mathbf{h}}$ be the solution output by LSV. According to Lemma C.7, $\tilde{\mathbf{h}} \in B_2(P_{y^\star}^{-\varepsilon_2}, +\varepsilon') \subseteq B_2(P_{y^\star}, -\varepsilon')$, which proves **(P2)**. For **(P1)**, note that Lemma C.7 also guarantees

$$V^\star - U_P(\tilde{\mathbf{h}}, y^\star) \leq \varepsilon' + \left(V^\star - \max_{\mathbf{h} \in P_{y^\star}^{-\varepsilon_2}} U_P(\mathbf{h}, y^\star)\right) \leq \varepsilon' + 2\varepsilon' = 3\varepsilon'.$$

Finally, we compute the total number of samples. By Lemma C.6, the initialization epoch takes $O\left(V^{-1}1.25^m r_\delta^{-1}(\frac{\eta}{k\sqrt{m}})\log^2 T\right)$ samples. According to Lemma C.2 and proposition C.3, each call to the APPROXMEM oracle requires $O\left(\sqrt{m}\left(\frac{\varepsilon_2}{\varepsilon_1}\right)^{\frac{m}{2}} r_\delta^{-1}(\frac{\varepsilon_1}{k\sqrt{m}})\log(T)\right) = O\left(\sqrt{m}\left(\frac{m}{\eta\varepsilon'}\right)^{\frac{m\gamma_0}{2}} r_\delta^{-1}(\frac{(\varepsilon'\eta)^{\gamma_0}}{km^{\gamma_0}\sqrt{m}})\log(T)\right)$ samples, and Lemma C.7 suggests that there LSV makes $O\left(m^2 \log^{O(1)}(\frac{m}{\varepsilon'\eta})\right)$ oracle calls, where we run $O(V^{-1}\log T)$ LSV instances for each initial point in the initialization set. Putting together, the total number of samples is

$$\tilde{O}\left(V^{-1}m^{\frac{5+m\gamma_0}{2}}(\eta\varepsilon')^{-\frac{m\gamma_0}{2}} r_\delta^{-1}(\frac{(\varepsilon'\eta)^{\gamma_0}}{km^{\gamma_0}\sqrt{m}\varepsilon})\right),$$

where $\tilde{O}$ hides logarithm terms in $T, m, 1/\eta, 1/\varepsilon'$. $\qquad\square$

## C.5 Representation length and post-processing

In this section, we show a post-processing of the output of LSV [42] under the representation length assumption.

**Assumption C.9** (Utility Representation Length). *Suppose the utility functions $U_P$ and $U_A$ are rational with denominator at most $a$, and normalized to be $[0,1]$. Therefore, the game's utility representation length is $L = 2mn\log a$.*

The next lemma shows that under Assumption C.9, the optimal solution $\mathbf{h}^\star$ also has a finite representation length.

**Lemma C.10.** *For any $y \in \mathcal{A}_A$, let $\mathbf{h}_y^\star \in \mathbb{R}^m$ be the principal's optimal strategy in $P_y$ that achieves $\max_{\mathbf{h}^\star \in P_y} U_P(\mathbf{h}^\star, y)$. Suppose the utility functions $U_P, U_A$ satisfy Assumption C.9. Then for all $i \in \mathcal{A}_P$, the $i$-th coordinate of $\mathbf{h}_y^\star$ is a rational number with denominator at most $2^{2mL}$.*

*Proof.* The proof of this lemma is similar to [12, Lemma 10], which follows from well-known results in linear programming. We first note that $\mathbf{h}^\star$ is the solution of the linear programming:

$$\max_{\mathbf{h}} U_P(\mathbf{h}, y), \quad \text{subject to } \mathbf{h}^\star \in P_y,$$

where $U_P(\mathbf{h}, y)$ is linear in $\mathbf{h}$ and the best response polytope $P_y$ can be represented as a system $\{\mathbf{h} : A\mathbf{h}^T \succeq \mathbf{b}\}$, where the set of constraints are the ones that define probability simplex $\mathcal{H}_P$, together with the constraints of the form $U_P(\mathbf{h}, y) - U_P(\mathbf{h}, y') \geq 0, \forall y' \neq y$. Suppose $A$ is normalized so that each entry is an integer. By Assumption C.9, each coefficient is at most $2^L$. Since the solution to the above LP is the intersection of $m$ independent constraints of $A$ (denoted with $D$), we have $\mathbf{h}_{y,i}^\star = \frac{\det(D_i)}{\det(D)}$ by Cramer's rule, where $D_i$ is $D$ with the $i$-th column replaced by $\mathbf{b}$. According to Hardamard's inequality,

$$\det(D) \leq \prod_{i\in[m]} \sqrt{\prod_{j\in[m]} d_{ij}^2} \leq \prod_{i\in[m]} \sqrt{m}2^L = m^{\frac{m}{2}}2^{Lm} \leq 2^{2Lm}.$$

$\qquad\square$

We also make the following assumption on the optimal strategy $\mathbf{h}^\star$: any near-optimal strategy $\tilde{\mathbf{h}}$ must lie in the neighborhood of $\mathbf{h}^\star$.

**Assumption C.11** (Near-optimal strategies). *There exists a constant $c$ s.t. for all $\varepsilon \leq 2^{-4mL}$ and all strategy $\tilde{\mathbf{h}}$ such that $U_P(\tilde{\mathbf{h}}.y^\star) \leq V^\star - \varepsilon$, we have $\|\tilde{\mathbf{h}} - \mathbf{h}^\star\|_\infty \leq c2^L\varepsilon$.*

Now we are ready to define the post-processing algorithm.

**Lemma C.12** (Post-processing). *Suppose that Assumptions C.9 and C.11 holds. Assume $\varepsilon \leq 2^{-4mL}$, $\tilde{\mathbf{h}}$ is an $\varepsilon$-approximate optimal strategy in $P_{\tilde{y}}$, and $\mathbf{h}_0$ is $\eta/2$-centered in $P_{\tilde{y}}$, i.e., $B_2(\mathbf{h}_0, \eta/2) \subseteq P_{\tilde{y}}$, then using no more than $O\left(2^{2m}r_\delta^{-1}(\frac{\lambda}{k\sqrt{m}})\log T\right)$ rounds of interactions with the agent, POSTPROCESS (Algorithm 5) outputs a strategy $\mathbf{h}$ that satisfies with probability $\geq 1 - \delta - T^{-1}$:*

---

**Algorithm 5:** Post-Processing (POSTPROCESS)

---

**Input:** non-robust strategy $(\tilde{\mathbf{h}}, \tilde{y})$ s.t. $\tilde{\mathbf{h}} \in B_2(P_{\tilde{y}}, \varepsilon)$, robustness parameter $\lambda$, intial point $\mathbf{h}_0$
**Output:** robust strategy $\mathbf{h}$
$S \leftarrow$ set of strategies $\mathbf{h}' \in \mathcal{H}_P$ s.t. each coordinate $\mathbf{h}'_i$ is a rational number with denominator at
  most $2^{2mL}$ and $|\tilde{\mathbf{h}}_i - \mathbf{h}_i| \leq c2^L\varepsilon$
**for** $\mathbf{h}^j \in S$ **do**
  $\mathbf{h}^j_{\text{query}} \leftarrow \left(1 - \frac{2\lambda}{\eta}\right) \mathbf{h}^j + \frac{2\lambda}{\eta} \mathbf{h}_0$
  Query APPROXMEM$_{\tilde{y}}(\mathbf{h}^j_{\text{query}})$ with $\varepsilon_1 = \varepsilon_2 = \frac{\lambda}{2}, \varepsilon_3 = \delta + T^{-2}$
  **if** APPROXMEM *returns* TRUE **then**
    $\mathbf{h} \leftarrow \left(1 - \frac{2\lambda}{\eta}\right) \mathbf{h}^j_{\text{query}} + \frac{2\lambda}{\eta} \mathbf{h}_0$
    RETURN $\mathbf{h}$

---

(i) $\mathbf{h}$ *lies robustly inside* $P_{\tilde{y}}$: $\mathbf{h} \in B_2(P_{\tilde{y}}, -\lambda)$;

(ii) $\mathbf{h}$ *is close to* $\tilde{\mathbf{h}}$: $\|\mathbf{h} - \tilde{\mathbf{h}}\|_2 \leq \sqrt{m}\left(c2^L\varepsilon + \frac{4\lambda}{\eta}\right)$.

*Proof.* To prove the lemma, we make the following claim: For all $\mathbf{h}_1 \in \mathcal{H}_P$, if $\mathbf{h}_1 \in P_{\tilde{y}}$ and $\mathbf{h}_2 = \left(1 - \frac{2\lambda}{\eta}\right)\mathbf{h}_1 + \frac{2\lambda}{\eta}\mathbf{h}_0$, then we have $\mathbf{h}_2 \in B_2(P_{\tilde{y}}, -\lambda)$. In fact, for all unit vectors $\mathbf{s} \in \mathbb{R}^m$, we have

$$\mathbf{h}_2 + \lambda\mathbf{s} = \left(1 - \frac{2\lambda}{\eta}\right)\mathbf{h}_1 + \frac{2\lambda}{\eta}\left(\mathbf{h}_0 + \frac{\eta}{2}\mathbf{s}\right).$$

Since $\mathbf{h}_1 \in P_{\tilde{y}}$ and $\mathbf{h}_0 + \frac{\eta}{2}\mathbf{s} \in B_2(\mathbf{h}_0, \eta/2) \subseteq P_{\tilde{y}}$ from the assumption that $\mathbf{h}_0$ is $\eta/2$-centered, their convex combination must also lie in $P_{\tilde{y}}$ because $P_{\tilde{y}}$ is convex. Thus, we have $\mathbf{h}_2 + \lambda\mathbf{s} \in P_{\tilde{y}}$. Since the above holds for all unit vectors $\mathbf{s}$, we conclude that $B_2(\mathbf{h}_2, \lambda) \subseteq P_{\tilde{y}}$. We have thus proved the claim.

With the above claim, we first show that when all oracle calls return correctly, $S$ contains at least one point $\mathbf{h}^j$ such that APPROXMEM returns TRUE on $\mathbf{h}^j_{\text{query}}$, and the post-processed version of this strategy satisfies (i) and (ii).

Since $\tilde{\mathbf{h}}$ is an $\varepsilon$-optimal strategy, by Assumption C.11, we know that the true optimal $\mathbf{h}^\star_{\tilde{y}}$ satisfies $\|\mathbf{h}^\star_{\tilde{y}} - \tilde{\mathbf{h}}\|_\infty \leq c2^L\varepsilon$. According to Lemma C.10, every coordinate of $\mathbf{h}^\star_{\tilde{y}}$ is a rational number with denominator at most $2^{2mL}$. Combining the two guarantees, we know that $S$ must contains the true optimal $\mathbf{h}^\star_{\tilde{y}}$, which satisfies $\mathbf{h}^\star_{\tilde{y}} \in P_{\tilde{y}}$. Suppose this point is $\mathbf{h}^j$. According to the above claim, we have $\mathbf{h}^j_{\text{query}} \in B_2(P_{\tilde{y}}, -\lambda)$. Since we simulated the APPROXMEM oracle with $\varepsilon_1 = \varepsilon_2 = \frac{\lambda}{2}$, it must return TRUE on strategies that belong to $B_2(P_{\tilde{y}}^{-\varepsilon_2}, -\varepsilon_1) = B_2(P_y, -\lambda)$, therefore, it must return TRUE on $\mathbf{h}^j_{\text{query}}$. This argument shows that POSTPROCESS always return valid strategies.

However, it could be the case that APPROXMEM returns TRUE on strategies $h^j$ that are not $\mathbf{h}^\star_{\tilde{y}}$. We argue that the guarantee of POSTPROCESS is unaffected by showing that if APPROXMEM returns TRUE on any $\mathbf{h}^j_{\text{query}}$, then we must have $\mathbf{h}^j_{\text{query}} \in B_2(P_{\tilde{y}}^{-\varepsilon_2}, +\varepsilon_1) \subseteq P_{\tilde{y}}$ from the guarantee of APPROXMEM (see Lemma C.2). Applying the above claim again, we have that $\mathbf{h} \in B_2(P_{\tilde{y}}, -\lambda)$ for the returned strategy $\mathbf{h}$, which proves (i).

As for (ii), note that

$$
\begin{aligned}
\|\mathbf{h} - \tilde{\mathbf{h}}\|_2 &\leq \|\tilde{\mathbf{h}} - \mathbf{h}^j\|_2 + \|\mathbf{h}^j - \mathbf{h}^j_{\text{query}}\|_2 + \|\mathbf{h}^j_{\text{query}} - \mathbf{h}\|_2 \\
&\leq \sqrt{m}\|\tilde{\mathbf{h}} - \mathbf{h}^j\|_\infty + \frac{2\lambda}{\eta}\left(\|\mathbf{h}^j - \mathbf{h}_0\|_2 + \|\mathbf{h}^j_{\text{query}} - \mathbf{h}_0\|_2\right) \\
&\leq c\sqrt{m}2^L\varepsilon + \frac{4\lambda\sqrt{m}}{\eta}.
\end{aligned}
$$

Finally, we analyze the failure probability and sample complexity of POSTPROCESS. Since the agent's adaptive calibration error is uniformly bounded by $r_\delta(\cdot)$ with probability $1 - \delta$, and the

probability that there exists an incorrect answer from APPROXMEM is bounded by $T \cdot T^{-2} = T^{-1}$ conditioned on agent's calibration error being bounded, the failure probability of POSTPROCESS is at most $\delta + T^{-1}$.

For the sample complexity, since $|S| \leq 2^m$ and each $\mathbf{h}^j \in S$ requires simulating APPROXMEM$_{\tilde{y}}$ with $O\left(1.25^m r_\delta^{-1}(\frac{\lambda}{k\sqrt{m}}) \log T\right)$ samples (see Lemma C.2 and proposition C.3), the total number of samples required is $O\left(2^{2m} r_\delta^{-1}(\frac{\lambda}{k\sqrt{m}}) \log T\right)$.

$\square$

**Theorem C.13.** *Suppose the utility functions satisfy Assumptions C.9 and C.11. If the principal uses Algorithm 2 with parameters $\varepsilon_1 = \sqrt{m}\varepsilon_2 = \left(\frac{\eta}{m2^{4mL}}\right)^{\gamma_0}, \varepsilon_3 = \delta + T^{-2}$ for the* APPROXMEM *oracle and $\varepsilon' = 2^{-4mL-1}$ for* LSV, *then feed the output of* LSV *into* POSTPROCESS$(\lambda)$*, then with probability at least $1 - \delta - T^{-1} - \varepsilon'$ the final solution satisfies the following two properties:*

**(P1)** $U_P(\tilde{\mathbf{h}}, y^\star) \geq V^\star - \frac{5m\lambda}{\eta}$*, i.e., $\tilde{\mathbf{h}}$ is an approximate Stackelberg equilibrium.*

**(P2)** $\tilde{\mathbf{h}} \in B_2(P_{y^\star}, -\lambda)$*, i.e., $\tilde{\mathbf{h}}$ lies robustly within the best-response polytope for $y^\star$.*

*The total number of samples needed is expressed in Equation (14).*

*Proof.* The analysis for the failure probability is the same with Theorem C.8 .

Now we consider the optimization problem for the optimal polytope $P_{y^\star}$, with initial point $\mathbf{h}_0$ that is $\frac{\eta}{2}$-centered in $P_{y^\star}$. Let $\hat{\mathbf{h}}$ be the solution output by LSV, and let $\tilde{\mathbf{h}}$ be the solution output by running POSTPROCESS on $\hat{\mathbf{h}}$. According to Lemma C.7, $\hat{\mathbf{h}}$ satisfies $\hat{\mathbf{h}} \in B_2(P_{y^\star}^{-\varepsilon_2}, +\varepsilon') \subseteq B_2(P_{y^\star}, \varepsilon')$, and the suboptimality of $\hat{\mathbf{h}}$ is at most $O(\varepsilon' + \varepsilon_2) \leq 2^{-4mL}$. Since $\varepsilon' \leq 2^{-4mL}$, by Lemma C.12, $\tilde{\mathbf{h}} \in B_2(P_{y^\star}, -\lambda)$, which proves **(P2)**.

To prove **(P1)**, note that since $\|\hat{\mathbf{h}} - \tilde{\mathbf{h}}\|_2 \leq \sqrt{m}\left(c2^L\varepsilon' + \frac{4\lambda}{\eta}\right)$ by Lemma C.12, we have

$$V^\star - U_P(\tilde{\mathbf{h}}, y^\star) \leq \left(V^\star - U_P(\hat{\mathbf{h}}, y^\star)\right) + \left(U_P(\hat{\mathbf{h}}, y^\star) - U_P(\tilde{\mathbf{h}}, y^\star)\right)$$
$$\leq 2^{-4mL} + m\left(c2^L\varepsilon' + \frac{4\lambda}{\eta}\right) \leq 2^{-3mL} + \frac{4m\lambda}{\eta} \leq \frac{5m\lambda}{\eta}.$$

Finally, we compute the total number of samples. According to Lemma C.2 and proposition C.3, each call to the APPROXMEM oracle requires $O\left(\sqrt{m}r_\delta^{-1}(\frac{\varepsilon_2}{k\sqrt{m}}) \log T\right) = O\left(\sqrt{m}r_\delta^{-1}(\frac{1}{km}\left(\frac{\eta}{m2^{4mL}}\right)^{\gamma_0}) \log T\right)$ samples, and Lemma C.7 suggests that LSV makes $O\left(m^2 \log^{O(1)}(\frac{m}{\varepsilon'\eta})\right) = O\left(m^3 L \log^{O(1)}(\frac{m}{\eta})\right)$ oracle calls, where we run $O(V^{-1}\log T)$ LSV instances for each initial point in the initialization set. In addition, POSTPROCESS takes $O\left(2^{2m}r_\delta^{-1}(\frac{\lambda}{k\sqrt{m}}) \log T\right)$ samples (Lemma C.12), and initialization takes $O\left(V^{-1}1.25^m r_\delta^{-1}(\frac{\eta}{k\sqrt{m}}) \log^2 T\right)$ samples (Lemma C.6). Adding them all up, the total number of samples is

$$\tilde{O}\left(V^{-1}1.25^m r_\delta^{-1}(\frac{\eta}{k\sqrt{m}}) + V^{-1}m^{3.5}Lr_\delta^{-1}(\frac{1}{km}\left(\frac{\eta}{m2^{4mL}}\right)^{\gamma_0}) + 2^{2m}r_\delta^{-1}(\frac{\lambda}{k\sqrt{m}})\right). \quad (14)$$

where $\tilde{O}$ hildes logarithm factors in $T, m, \eta^{-1}$.

$\square$

## C.6  Proof of Theorem 3.2

In this section, we present the main theorem (Theorem 3.2) in Section 3. We first formally restate the theorem with exact convergence rate.

**Theorem C.14** (Formal version of Theorem 3.2). *There exists an algorithm for the principal in CSGs that achieves average utility with probability $1 - o(1)$:*

$$\lim_{T \to \infty} \frac{1}{T} \sum_{t \in [T]} U_P(\mathbf{h}_t, y_t) \geq V^\star. \tag{15}$$

*Specifically, for calibrated agents with calibration error rate $r_\delta(t) = O(t^{-1/\beta})$,*

*(I) If the principal runs the* EXPLORE-THEN-COMMIT *algorithm (Algorithm 1) where the explore phase follows the parameter settings in Theorem C.8 with $\varepsilon' = O\left(T^{-\frac{1}{\gamma_1}} C_1^{\frac{1}{\gamma_1}} \eta^{-\frac{\gamma_1 - 1}{\gamma_1}}\right)$, then the limit in* (15) *approaches $V^\star$ with rate*

$$\tilde{O}\left(T^{-\frac{1}{\gamma_1}} C_1^{\frac{1}{\gamma_1}} \eta^{-\frac{\gamma_1 - 1}{\gamma_1}}\right),$$

*where*

$$C_1 \triangleq V^{-1} m^{\frac{5 + m\gamma_0 + \beta(2\gamma_0 + 1)}{2}} k^\beta, \quad \gamma_1 \triangleq \gamma_0 \left(\frac{m}{2} + \beta\right) + 1.$$

*(II) If the principal runs the* EXPLORE-THEN-COMMIT *algorithm (Algorithm 1) where the explore phase follows the parameter settings in Theorem C.13 with $\lambda = O\left(\left(\frac{\eta C_2}{m}\right)^{\frac{1}{\gamma_2}} T^{-\frac{1}{\gamma_2}}\right)$, then the limit in* (15) *approaches $V^\star$ with rate*

$$\tilde{O}\left(T^{-\frac{1}{\gamma_2}} \left(\frac{m}{\eta}\right)^{1 - \frac{1}{\gamma_2}} C_2^{\frac{1}{\gamma_2}}\right),$$

*where*

$$C_2 \triangleq V^{-1} 2^{5mL\beta\gamma_0} k^\beta, \quad \gamma_2 \triangleq \beta\gamma_0 + 1.$$

*Proof.* According to Lemmas C.4 and C.5, we have that with probability $1 - o(1)$, Algorithm 2 (with potential post-processing) returns a policy $\tilde{\mathbf{h}}$ within $N$ samples, where $\tilde{\mathbf{h}}$ satisfies

**(P1)** $U_P(\tilde{\mathbf{h}}, y^\star) \geq V^\star - \varepsilon_{\text{opt}}$, i.e., $\tilde{\mathbf{h}}$ is an approximate Stackelberg equilibrium.

**(P2)** $\tilde{\mathbf{h}} \in B_2(P_{y^\star}, -\varepsilon_{\text{robust}})$, i.e., $\tilde{\mathbf{h}}$ lies *robustly* within the best-response polytope for $y^\star$.

With properties **(P1)**and **(P2)**, the arguments in Section 3 give us

$$V^\star T - \sum_{t \in [T]} U_P(\mathbf{h}_t, y_t) \lesssim N + \varepsilon_{\text{opt}} T + \frac{k\sqrt{m} T r_\delta(T)}{\varepsilon_{\text{robust}}}.$$

Now we plug in specific instantiations of Algorithm 1.

(I) Suppose the principal follows the parameter settings in Theorem C.8. In this case, we have $\varepsilon_{\text{opt}} = 3\varepsilon'$, $\varepsilon_{\text{robust}} = \varepsilon'$, and

$$N = \tilde{O}\left(V^{-1} m^{\frac{5 + m\gamma_0}{2}} (\eta\varepsilon')^{-\frac{m\gamma_0}{2}} r_\delta^{-1}(\frac{(\varepsilon'\eta)^{\gamma_0}}{km^{\gamma_0}\sqrt{m}})\right)$$
$$= \tilde{O}\left(V^{-1} m^{\frac{5 + m\gamma_0 + \beta(2\gamma_0 + 1)}{2}} k^\beta (\eta\varepsilon')^{-\gamma_0\left(\frac{m}{2} + \beta\right)}\right).$$

Finally, optimizing over $\varepsilon'$ gives the claimed bound.

(II) Suppose the principal follows the parameter settings in Theorem C.13. In this case, we have $\varepsilon_{\text{opt}} = \frac{5m\lambda}{\eta}$, $\varepsilon_{\text{robust}} = \lambda$, and

$$N = \tilde{O}\left(V^{-1} 1.25^m r_\delta^{-1}(\frac{\eta}{k\sqrt{m}}) + V^{-1} m^{3.5} L r_\delta^{-1}(\frac{1}{km}\left(\frac{\eta}{m2^{4mL}}\right)^{\gamma_0}) + 2^{2m} r_\delta^{-1}(\frac{\lambda}{k\sqrt{m}})\right)$$
$$= \tilde{O}\left(V^{-1} 1.25^m (k\sqrt{m})^\beta \eta^{-\beta} + V^{-1} 2^{5mL\beta\gamma_0} k^\beta \lambda^{-\beta\gamma_0}\right).$$

Finally, optimizing over $\lambda$ gives the claimed bound.

**Remark C.15** (Representation length). *If we assume the optimal solution $\mathbf{h}^\star$ lies in the grid of size $poly(\varepsilon^{-1})$, then in case (II) of Theorem C.14, we have $2^{2mL} = poly(\varepsilon^{-1})$, which leads to a rate of $\tilde{O}\left(poly(\varepsilon, k, m, \eta^{-1}, V^{-1}) \cdot T^{-\frac{1}{\gamma_2}}\right)$, where $\gamma_2$ is a constant that only depends on the agent's calibration error rate $\beta$. This is the result of replacing Lemma C.10 with the finite grid assumption.*

**Remark C.16** (Adaptive regret versus calibration). *Our primary focus lies on calibration due to its characterization of agents' beliefs and the fact that it provides both upper and lower bounds to the principal's utility. This is particularly useful for the learning direction, as denoted by the lower bounds in Theorems 3.2 and C.14. However, a different form of adaptive guarantee would suffice here: one concerning (external) regret. Nevertheless, we do not focus on regret as a characterization as it doesn't offer the same upper bound guarantees — in fact, the principal could potentially extract more utility than $V^\star$. Additionally, regret-based assumptions tend to overly emphasize the agent's optimization techniques rather than maintaining a consistent belief about the action being executed.*

$\square$

# D    Supplementary Material for Section 4

## D.1    Background on Sleeping Experts and ADANORMALHEDGE

We start the exposition of this part by introducing the sleeping experts problem [10, 31]. For each expert $i \in [N]$ and round $t \in [T]$, let $\ell_{t,i} \in [0,1]$ be the loss of expert $i$, and let $I_{t,i}$ be an indicator that takes value $I_{t,i} = 1$ if expert $i$ is active at round $t$ and $I_{t,i} = 0$ if asleep. The interaction protocol at each round $t$ goes as follows: The indicators $(I_{t,i})_{i \in [N]}$ are revealed to the learner. The learner selects a probability distribution $\pi_t \in \Delta([N])$ that is supported only on the set of active experts $A_t \triangleq \{i : I_{t,i} = 1\}$. The adversary selects a loss vector $(\ell_{t,i})_{i \in [N]}$. The learner then suffers expected loss $\hat{\ell}_t = \mathbb{E}_{i \sim \pi_t}[\ell_{t,i}]$. The regret with respect to each expert $i$ only accounts for the rounds when $i$ is awake, which, together with the fact that $\pi_t$ is only supported on active experts, implies that

$$\text{Reg}_T(i) = \sum_{t \in [T]} I_{t,i} \left( \hat{\ell}_t - \ell_{t,i} \right) \quad \Rightarrow \quad \text{Reg}_T = \max_i \text{Reg}_T(i) \tag{16}$$

One of the algorithms that can be used to provide sublinear regret for the sleeping experts problem is ADANORMALHEDGE [44]. ADANORMALHEDGE is a powerful, parameter-free algorithm which provides regret bounds in terms of the cumulative magnitude of the *instantaneous* regrets, defined as: $r_{t,i} = \hat{\ell}_t - \ell_{t,i}$ for all experts $i \in [N]$. As its name suggests, ADANORMALHEDGE uses the well-known algorithm HEDGE as a backbone; HEDGE maintains a probability distribution over experts at each round $t$ and draws an expert from said distribution. After the expert's loss is revealed, the probability distribution for the next round $t+1$ is updated using a multiplicative weights argument. For bandit feedback (i.e., when only the chosen expert's loss is revealed to the learner), the multiplicative weights update rule uses an inverse propensity scoring estimator for each expert's loss in place of their real loss. The new element that ADANORMALHEDGE brings to the table is a way of defining the weights at each round $t$; specifically, the weights are updated proportionally to the sum of instantaneous regret for each expert until round $t$. This allows the learner to obtain finer control over the total regret without needing extra parameters to tune the algorithm at each round. The exact regret guarantee that ADANORMALHEDGE obtains is stated formally below.

**Lemma D.1** (ADANORMALHEDGE [44]). *Let $r_{t,i} = I_{t,i}\left(\hat{\ell}_t - \ell_{t,i}\right)$ be the instantaneous regret of any active expert $i \in A_t$ at round $t$, and $c_{t,i} = |r_{t,i}|$. Then, ADANORMALHEDGE with prior $q \in \Delta([N])$ selects experts according to the following distribution*

$$\pi_{t,i} \propto q_i I_{t,i} w(R_{t-1,i}, C_{t-1,i}), \text{ where}$$

$$R_{t-1,i} = \sum_{\tau \in [t-1]} r_{\tau,i}, \; C_{t-1,i} = \sum_{\tau \in [t-1]} c_{\tau,i},$$

$$w(R,C) = \frac{1}{2}\left(\Phi(R+1, C+1) - \Phi(R-1, C+1)\right),$$

$$\Phi(R,C) = \exp\left(\frac{\max\{0, R\}^2}{3C}\right)$$

*The regret of* ADANORMALHEDGE *against any distribution over experts* $u \in \Delta([N])$ *is bounded by*

$$\text{Reg}_T(u) \leq O\left(\sqrt{\langle u, C_T \rangle \cdot (D_{KL}(u\|q) + \log \log T + \log \log N))}\right).$$

*where by* $D_{KL}(u\|q)$ *we denote the KL-divergence between distributions* $u$ *and* $q$.

ADANORMALHEDGE can be used to obtain adaptive regret bounds by creating a sleeping expert $(i, s)$ for each $i \in [N], s \in [T]$ that has the same loss as expert $i$ but is only awake after $s$.

**Corollary D.2.** *Running* ADANORMALHEDGE *for the sleeping expert setting with prior* $q_{(i,s)} \propto \frac{1}{s^2}$ *gives regret*

$$\text{Reg}_t((i, s)) \leq O\left(\sqrt{(t - s)\left(\log(Ns) + \log \log T\right)}\right),$$

*where* $T_i = \sum_{t=1}^{T} I_{t,i}$ *is the total number of rounds in which* $i$ *is active.*

### D.2 Formula for Computing $Q_t$ when $m = 2$

To obtain the explicit formula for $Q_t$, we first discretize the space of forecasts $\mathcal{F}_P = [0, 1]$ (since we focus on the case where $m = 2$) to form set $\mathcal{F}_P^\varepsilon = \{0, \varepsilon, 2\varepsilon, \dots, 1 - \varepsilon, 1\}$. Then, we have that for each $\hat{\mathbf{p}} \in \mathcal{F}_P^\varepsilon$:

$$\mathbb{E}_{g \sim \pi_t} [\ell_{t,g}] = \mathbb{E}_{g \sim \pi_t} [L_g(\mathbf{h}_t, \mathbf{p})] = \sum_{g \in A_t(\mathcal{G})} \pi_{t,g} w_i(\mathbf{p}) \sigma(\mathbf{h}_t - \mathbf{p})$$

$$= (\mathbf{h}_t - \mathbf{p}) \underbrace{\sum_{i \in \mathcal{A}_A} w_i(\mathbf{p}) \sum_{s \leq t} \left(\pi_{t,g(s,i,+1)} - \pi_{t,g(s,i,-1)}\right)}_{Z_{\mathbf{p}}} \qquad (17)$$

where we have omitted index $j$ from the sleeping expert $g$ since because $m = 2$, we can focus on a single action $j$. We assume WLOG that no forecast $\mathbf{p} \in \mathcal{F}_P^\varepsilon$ falls exactly on the boundary of best response polytopes, so there is no tie-breaking needed. From Equation (17), we have that:

$$\max_{\mathbf{h}_t \in \mathcal{H}_P} \mathbb{E}_{g \sim \pi_t} [\ell_{t,g}] = \max\{Z_{\mathbf{p}}, 0\} - \mathbf{p} Z_{\mathbf{p}} \qquad (18)$$

where the equation also uses the fact that for $m = 2$, $\max_{\mathbf{h}_t \in \mathcal{H}_P} \mathbf{h}_t = 1$.

In the final step, we map $\mathbf{p}$ to the discretized grid of $\mathcal{F}_P^\varepsilon$. Let $j\varepsilon, (j + 1)\varepsilon$ be two adjacent discretized points and $q \in [0, 1]$ such that: $qZ_{j\varepsilon} + (1 - q)Z_{(j+1)\varepsilon}0$. Then, setting $q_{t,j\varepsilon} = q$ and $q_{t,(j+1)\varepsilon} = 1 - q$ and using Equation (18) gives that

$$\max_{\mathbf{h}_t \in \mathcal{H}_P} \mathbb{E}_{\substack{g \sim \pi_t \\ \mathbf{p} \sim Q_t}} [\ell_{t,g}] \leq \varepsilon.$$

### D.3 Proof of Theorem 4.1

*Proof of Theorem 4.1.* We first specify how to build a set of sleeping experts settings from our problem definition. For that, consider the following set of experts:

$$\mathcal{G} = \left\{g_{(s,i,j,\sigma)} : s \in [T], i \in \mathcal{A}_A, j \in \mathcal{A}_P, \sigma \in \{\pm 1\}\right\},$$

i.e., we create a different expert for each round, each principal-agent action pair, and each $\sigma$ (the use of which will be made clear in the next paragraph). For each expert $g_{(s,i,j,(\sigma)} \in \mathcal{G}$ and $t \in [T]$, we define the loss, sleeping/awake indicator, and instantaneous regret respectively as:

$$\ell_{t,g_{(s,i,j,\sigma)}} \triangleq L_{g_{(i,i,j,\sigma)}}(\mathbf{h}_t, \mathbf{p}_t) = w_i(\mathbf{p}_t) \cdot \sigma \cdot (h_{t,j} - p_{t,j});$$

$$I_{t,g_{(s,i,j,\sigma)}} \triangleq \mathbf{1}\{t \geq s\};$$

$$r_{t,g} \triangleq I_{t,g} \cdot \left(\ell_{t,g} - \hat{\ell}_t\right).$$

where by $h_{t,j}, p_{t,j}$ we denote the $j$-th coordinate of the $\mathbf{h}$ and $\mathbf{p}$ vectors respectively.

Running ADANORMALHEDGE on the instance with $\mathcal{G}$ that we specified above, with prior $q_{g_{(s,i,j,\sigma)}} \propto \frac{1}{s^2}$ [44, Section 5.1] guarantees that $\forall g_{(s,i,j,\sigma)} \in \mathcal{G}$,

$$\text{Reg}_t \left( g_{(s,i,j,\sigma)} \right) = \sum_{\tau \in [t]} r_{t,g_{(s,i,\sigma)}} \leq \widetilde{O}\left( \sqrt{(t-s)\log(kmT)} \right). \tag{19}$$

where $\tilde{O}(\cdot)$ hides lower order poly-logarithmic terms. Therefore, by simulating the NRBR dynamics, we obtain that with probability at least $\geq 1 - \delta$, $\forall i \in [k]$ and $1 \leq s < t \leq T$,

$$\text{CalErr}_i(\mathbf{h}_{s:t}, \mathbf{p}_{s:t}) = \frac{1}{t-s} \max_{j \in \mathcal{A}_P} \max_{\sigma \in \{\pm 1\}} \sum_{\tau=s}^{t} I_{\tau,g_{(s,i,j,\sigma)}} \cdot \ell_{\tau,g_{(s,i,j,\sigma)}} \qquad \text{(Definition 2.3)}$$

$$= \frac{1}{t-s} \max_{j \in \mathcal{A}_P} \max_{\sigma \in \{\pm 1\}} \underbrace{\sum_{\tau=s}^{t} I_{\tau,g_{(s,i,j,\sigma)}} \cdot \left( \ell_{\tau,g_{(s,i,j,\sigma)}} - \mathop{\mathbb{E}}_{g \sim \pi_\tau} [\ell_{\tau,g}] \right)}_{\text{Reg}_t \left( g_{(s,i,j,\sigma)} \right)}$$

$$+ \frac{1}{t-s} \max_{j \in \mathcal{A}_P} \max_{\sigma \in \{\pm 1\}} \sum_{\tau=s}^{t} I_{\tau,g_{(s,i,j,\sigma)}} \cdot \mathop{\mathbb{E}}_{g \sim \pi_\tau} [\ell_{\tau,g}]$$

$$= \frac{1}{t-s} \max_{j \in \mathcal{A}_P} \max_{\sigma \in \{\pm 1\}} \text{Reg}_t \left( g_{(s,i,j,\sigma)} \right) + \frac{1}{t-s} \max_{j \in \mathcal{A}_P} \max_{\sigma \in \{\pm 1\}} \underbrace{\sum_{\tau=s}^{t} \mathop{\mathbb{E}}_{g \sim \pi_\tau} \left[ L_{g_{(s,i,j,\sigma)}}(\mathbf{h}_\tau, \mathbf{p}_\tau) \right]}_{\mathcal{T}} \tag{20}$$

where for the first derivation we add and subtract $\sum_\tau \hat{\ell}_\tau$ and use that because of the NRBR dynamics: $\hat{\ell}_\tau = \mathbb{E}_{g \sim P_\tau} [\ell_{\tau,g}]$, and for the last derivation we have used the definition of $\ell_{\tau,g} = L_g(\mathbf{h}_\tau, \mathbf{p}_\tau)$ (Equation (6)). We next upper bound the term $\mathcal{T}$ as follows:

$$\mathcal{T} = \sum_{\tau=s}^{t} \mathop{\mathbb{E}}_{\substack{g \sim \pi_\tau \\ \mathbf{p} \sim Q_\tau}} [L_g(\mathbf{h}_\tau, \mathbf{p})] + \sum_{\tau=s}^{t} \mathop{\mathbb{E}}_{g \sim \pi_\tau} [L_g(\mathbf{h}_\tau, \mathbf{p}_\tau)] - \sum_{\tau=s}^{t} \mathop{\mathbb{E}}_{\substack{g \sim \pi_\tau \\ \mathbf{p} \sim Q_\tau}} [L_g(\mathbf{h}_\tau, \mathbf{p})]$$

$$\leq \sum_{\tau=s}^{t} \max_{\mathbf{h}_\tau \in \Delta(\mathcal{A}_P)} \mathop{\mathbb{E}}_{\substack{g \sim \pi_\tau \\ \mathbf{p} \sim Q_\tau}} [L_g(\mathbf{h}_\tau, \mathbf{p})] + \sum_{\tau=s}^{t} \mathop{\mathbb{E}}_{g \sim \pi_\tau} \left[ L_g(\mathbf{h}_\tau, \mathbf{p}_\tau) - \mathop{\mathbb{E}}_{\mathbf{p} \in Q_\tau} [L_g(\mathbf{h}_\tau, \mathbf{p})] \right]$$

$$\leq \varepsilon \cdot (t-s) + \sqrt{(t-s)\log((t-s)/\delta)}$$

where the first inequality is by the property of $\mathbf{h}_\tau$ being the best strategy for the principal, and the last one uses the fact that $\max_{\mathbf{h}_\tau \in \Delta(\mathcal{A}_P)} \mathbb{E}_{\substack{g \sim \pi_\tau \\ \mathbf{p} \sim Q_\tau}} [L_g(\mathbf{h}_\tau, \mathbf{p})] \leq \varepsilon$ from the NRBR Equation (8) and a martingale concentration bound on the second term.

Plugging the upper bound for $Q$ back to Equation (20) and using the regret bound for AdaNormal-Hedge (Equation (19)) we get:

$$\text{CalErr}_i(\mathbf{h}_{s:t}, \mathbf{p}_{s:t}) \leq \widetilde{O}\left( \frac{\log(kmT)}{t-s} \right) + O\left( \frac{\log((t-s)/\delta)}{t-s} \right) + \varepsilon \leq r_t(\delta) + \epsilon.$$

$\square$

# E   Supplementary Material for Section 5

## E.1   Proof of Theorem 5.2

**Theorem 5.2** (Restated). *For continuous CSGs satisfying Assumption 5.1, for all $\varepsilon_0 > 0$, there exists a finite binning $\Pi_0$ such that if the agent is $(0, \Pi_0)$ - adaptively calibrated and the principal runs an appropriately parametrized instance of* LAZYGDWOG *(Algorithm 3) then:*

$$\lim_{\substack{\Phi \to \infty \\ M \to \infty}} \frac{1}{\Phi M} \sum_{\phi \in [\Phi]} \sum_{i \in [M]} U_P(\mathbf{h}_\phi, y_{\phi,i}) \geq V^\star - \varepsilon_0.$$

*Moreover, for any sequence of the principal's actions $\mathbf{h}_{[1:T]}$, it holds that:*

$$\lim_{T \to \infty} \frac{1}{T} \sum_{t \in [T]} U_P(\mathbf{h}_t, y_t) \leq V^\star + \varepsilon_0.$$

### E.1.1   Proof of Lower Bound

*Proof.* Before delving into the proof of the lower bound, we first introduce some notations. Let $C(\mathbf{h}) \triangleq U_P(\mathbf{h}, \mathrm{BR}(\mathbf{h}))$. Let $V_\delta^\star \triangleq \max_{\mathbf{h} \in B_2(\mathcal{A}_P, -\delta)} C(\mathbf{h})$ be the optimal utility restricted in the smaller strategy set $B_2(\mathcal{A}_P, -\delta)$. We use $\overline{y}_\phi \triangleq \frac{1}{M} \sum_{s \in [M]} y_{\phi, s}$ to denote the average feedback that LazyGDwoG uses to update the strategies.

We first consider any fixed $\varepsilon > 0$. Combining the guarantees of Lemmas E.4 and E.5, we conclude that there exists a finite binning $\Pi_0$ and $M_\varepsilon < \infty$, such that if the agent is $(0, \Pi_0)$-adaptively calibrated, then $\forall M \geq M_\varepsilon$, the following two inequalities are satisfied at the same time:

$$\sup_{\phi \in [\Phi]} \left\| \overline{y}_\phi - \mathrm{BR}(\mathbf{h}_\phi) \right\|_2 \leq \varepsilon \qquad \text{(by Lemma E.4)} \qquad (21)$$

$$\sup_{\phi \in \Phi} \frac{1}{M} \sum_{s \in [M]} U_P(\mathbf{h}_\phi, y_{\phi, s}) \geq C(\mathbf{h}_\phi) - \varepsilon; \qquad \text{(by Lemma E.5)} \qquad (22)$$

Set the parameters according to $\gamma_\phi = \gamma_0 m^{-\frac{1}{2}} \phi^{-\frac{3}{4}}$ and $\delta_\phi \equiv \delta = \delta_0 m^{\frac{1}{2}} \Phi^{-\frac{1}{4}}$ in Algorithm 3, then similar arguments to [51, Theorem 3.1] guarantee that

$$V_\delta^\star - \frac{1}{\Phi} \sum_{\phi \in [\Phi]} \mathbb{E}[C(\mathbf{h}_\phi)] \leq \left( \frac{D_P^2}{2\gamma_0} + \frac{2W_P^2}{\delta_0^2} \right) \sqrt{m} \Phi^{-\frac{1}{4}} + L_{\mathrm{BR}} D_P \frac{1}{\Phi} \sum_{\phi \in [\Phi]} \| \overline{y}_\phi - \mathrm{BR}(\mathbf{h}_\phi) \|_2$$

$$\overset{(a)}{\leq} \left( \frac{D_P^2}{2\gamma_0} + \frac{2W_P^2}{\delta_0^2} \right) \sqrt{m} \Phi^{-\frac{1}{4}} + L_{\mathrm{BR}} D_P \cdot \varepsilon,$$

where (a) is from Equation (21).

Now we upper bound the difference between $V^\star$ and $V_\delta^\star = \max_{\mathbf{h} \in B_2(\mathcal{A}_P, -\delta)} C(\mathbf{h})$, then we have

$$V^\star - V_\delta^\star \leq \max_{\mathbf{h}^\star \in \mathcal{A}_P} \min_{\mathbf{h}' \in B_2(\mathcal{A}_P, -\delta)} C(\mathbf{h}^\star) - C(\mathbf{h}') \leq L_U \max_{\mathbf{h}^\star \in \mathcal{A}_P} \min_{\mathbf{h}' \in B_2(\mathcal{A}_P, -\delta)} \| \mathbf{h}^\star - \mathbf{h}' \|_2 \leq L_U \delta,$$

where the second inequality follows from Assumption 5.1 that $C(\mathbf{h})$ is $L_U$-Lipschitz.

The next step is to upper bound the difference between the actual average utility and $\frac{1}{\Phi} \sum_{\phi \in [\Phi]} \mathbb{E}[C(\mathbf{h}_\phi)]$. From Equation (22), we have

$$\frac{1}{\Phi} \sum_{\phi \in [\Phi]} \mathbb{E}[C(\mathbf{h}_\phi)] - \frac{1}{\Phi M} \sum_{\phi \in [\Phi]} \sum_{i \in [M]} U_P(\mathbf{h}_\phi, y_{\phi, i}) \leq \varepsilon.$$

Finally, putting the above inequalities together, we obtain

$$V^\star - \frac{1}{\Phi M} \sum_{\phi \in [\Phi]} \sum_{i \in [M]} U_P(\mathbf{h}_\phi, y_{\phi, i})$$

$$\leq (V^\star - V_\delta^\star) + \left( V_\delta^\star - \frac{1}{\Phi} \sum_{\phi \in [\Phi]} \mathbb{E}[C(\mathbf{h}_\phi)] \right) + \left( \frac{1}{\Phi} \sum_{\phi \in [\Phi]} \mathbb{E}[C(\mathbf{h}_\phi)] - \frac{1}{\Phi M} \sum_{\phi \in [\Phi]} \sum_{i \in [M]} U_P(\mathbf{h}_\phi, y_{\phi, i}) \right)$$

$$\leq L_U \delta_0 m^{\frac{1}{2}} \Phi^{-\frac{1}{4}} + \left( \frac{D_P^2}{2\gamma_0} + \frac{2W_P^2}{\delta_0^2} \right) \sqrt{m} \Phi^{-\frac{1}{4}} + L_{\mathrm{BR}} D_P \cdot \varepsilon + \varepsilon.$$

Taking the limit of $\Phi \to \infty$, the above inequalities imply

$$\lim_{\substack{\Phi \to \infty \\ M \to \infty}} \frac{1}{\Phi M} \sum_{\phi \in [\Phi]} \sum_{i \in [M]} U_P(\mathbf{h}_\phi, y_{\phi, i}) \geq V^\star - \varepsilon \left( L_{\mathrm{BR}} D_P + 1 \right).$$

Since the above arguments hold for all $\varepsilon > 0$, taking $\varepsilon = \frac{\varepsilon_0}{L_{\mathrm{BR}} D_P + 1}$ proves the theorem. $\qquad \square$

### E.1.2 Proof of Upper Bound

For a fixed $\varepsilon > 0$, let $D_\varepsilon = \{x_1, \cdots, x_I\}$ be an $\varepsilon$-grid of $\mathcal{F}_P$ under $\ell_2$ distance, and let $\Pi_0$ be the continuous binning specified by Equation (26). We have:

$$
\begin{aligned}
\sum_{t \in [T]} U_P(\mathbf{h}_t, y_t) &= \sum_{i \in [I]} \sum_{t \in [T]} w_i(\mathbf{p}_t) U_P(\mathbf{h}_t, \mathrm{BR}(\mathbf{p}_t)) \\
&\overset{(a)}{\leq} \sum_{i \in [I]} \sum_{t \in [T]} w_i(\mathbf{p}_t) \left( U_P(\mathbf{h}_t, \mathrm{BR}(x_i)) + L_2 \cdot L_{\mathrm{BR}} \underbrace{\|\mathbf{p}_t - x_i\|_2}_{\leq 2\varepsilon} \right) \\
&\overset{(b)}{\leq} \sum_{i \in [I]} \left( \sum_{t \in [T]} w_i(\mathbf{p}_t) \right) U_P \Big( \frac{\sum_{t \in [T]} w_i(\mathbf{p}_t) \mathbf{h}_t}{\sum_{t \in [T]} w_i(\mathbf{p}_t)}, \mathrm{BR}(x_i) \Big) + 2 L_2 L_{\mathrm{BR}} \varepsilon T \\
&\overset{(c)}{=} \sum_{i \in [I]} n_T(i) U_P(\overline{\mathbf{h}}_T(i), \mathrm{BR}(x_i)) + 2 L_2 L_{\mathrm{BR}} \varepsilon T \\
&\overset{(d)}{\leq} \sum_{i \in [I]} n_T(i) \Big( U_P(\overline{\mathbf{p}}_T(i), \mathrm{BR}(x_i)) + L_1 \big\|\overline{\mathbf{p}}_T(i) - \overline{\mathbf{h}}_T(i)\big\|_2 \Big) + 2 L_2 L_{\mathrm{BR}} \varepsilon T \\
&= \underbrace{\sum_{i \in [I]} n_T(i) U_P(\overline{\mathbf{p}}_T(i), \mathrm{BR}(x_i))}_{(A)} + \underbrace{L_1 \sum_{i \in [I]} n_T(i) \big\|\overline{\mathbf{p}}_T(i) - \overline{\mathbf{h}}_T(i)\big\|_2}_{(B)} + 2 L_2 L_{\mathrm{BR}} \varepsilon T
\end{aligned}
\tag{23}
$$

In the above inequalities that lead to (23), step (a) is because $U_P$ is $L_2$-Lipschitz in the second argument and $\mathrm{BR}(\cdot)$ is $L_{\mathrm{BR}}$-Lipschitz, and the fact that $w_i(\mathbf{p}_t) > 0$ only when $\|\mathbf{p}_t - x_i\|_2 < 2\varepsilon$. In step (b), we used Jensen's inequality because $U_P$ in concave in the first argument. Step (c) follows from the definition of $n_T(i)$ and $\overline{\mathbf{h}}_T(i)$ in Definition 2.3. The last inequality (d) uses the fact that $U_P$ is $L_1$-Lipschitz in the first argument to decompose $U_P(\overline{\mathbf{p}}_T(i), \mathrm{BR}(x_i))$ into calibration error (i.e., term (B)) and $U_P(\overline{\mathbf{p}}_T(i), \mathrm{BR}(x_i))$ where the strategy that the agent best responds to is close to the principal's strategy (i.e., term (A)).

We can further bound $(A)$ and $(B)$ in Equation (23) respectively as follows:

$$
(A) \leq \sum_{i \in [I]} n_T(i) \left( U_P(x_i, \mathrm{BR}(x_i)) + L_1 \|x_i - \overline{\mathbf{p}}_T(i)\|_2 \right) \leq V^\star T + L_1(2\varepsilon) T,
$$

and

$$
(B) \leq L_1 T \sum_{i \in [I]} \mathrm{CalErr}_i(\mathbf{h}_{1:T}, \mathbf{p}_{1:T}) \leq L_1 |D_\varepsilon| r_\delta(T) T \quad \text{w.p.} \ \geq 1 - \delta.
$$

Therefore, putting the above bounds together, we obtain that with probability $\geq 1 - \delta$,

$$
\frac{1}{T} \sum_{t \in [T]} U_P(\mathbf{h}_t, y_t) \leq V^\star + (L_1 |D_\varepsilon|) r_\delta(T) + 2(L_1 + L_2 L_{\mathrm{BR}}) \varepsilon.
$$

Since the above derivation holds for any $\varepsilon > 0$, it suffices to take $\varepsilon$ such that $2(L_1 + L_2 L_{\mathrm{BR}})\varepsilon = \varepsilon_0$. Finally, since $|D_\varepsilon| < \infty$ and $r_\delta(T) = o(1)$, taking the limit of $T \to \infty$ proves the upper bound:

$$
\lim_{T \to \infty} \frac{1}{T} \sum_{t \in [T]} U_P(\mathbf{h}_t, y_t) \leq V^\star + \varepsilon_0.
$$

### E.2 Key lemma: asymptotically correct forecast

In this section, we state and prove the key lemma for establishing Theorem 5.2. Intuitively, this lemma states that for any strategy $\mathbf{h} \in \mathcal{A}_P$, as long as the principal repeatedly plays $\mathbf{h}$ for enough rounds, the fraction of times where the agent's forecast is close to $\mathbf{h}$ will converge to 1.

**Lemma E.1.** *For any $\varepsilon_0 > 0$, there exists a finite binning $\Pi_0$, such that if the principal repeatedly plays any $\mathbf{h} \in \mathcal{A}_P$ for $M$ rounds and the agent's forecasts $\mathbf{p}_{1:M}$ are $(0, \Pi_0)$- adaptively calibrated, then:*

$$\lim_{M \to \infty} \frac{1}{M} \left| \{ s \in [M] : \|\mathbf{p}_s - \mathbf{h}\|_2 \geq \varepsilon_0 \} \right| = 0 \tag{24}$$

*In particular, if the calibration error (defined in Definition 2.3) has rate $r(\cdot) \in o(1)$ with respect to $\Pi_0$, then*

$$\frac{1}{M} \left| \{ s \in [M] : \|\mathbf{p}_s - \mathbf{h}\|_2 \geq \varepsilon_0 \} \right| \leq \frac{8\sqrt{m}|\Pi_0|^2}{\varepsilon_0} r(M). \tag{25}$$

*Proof of Lemma E.1.* We first describe the construction of $\Pi_0$. For $\varepsilon = \frac{1}{4}\varepsilon_0$, let $D_\varepsilon = \{x_1, \cdots, x_I\}$ be an $\varepsilon$-grid of $\mathcal{F}_P$ under $\ell_2$ distance, and $\Lambda(\mathbf{p}; x, R) \triangleq (R - \|\mathbf{p} - x\|_2)_+$ be the tent function with center $x$ and radius $R$. Consider the following binning

$$\Pi_0 = \left\{ w_i(\mathbf{p}) \triangleq \frac{\Lambda(\mathbf{p}; x_i, 2\varepsilon)}{\sum_{j \in [I]} \Lambda(\mathbf{p}; x_j, 2\varepsilon)} : x_i \in D_\varepsilon \right\}. \tag{26}$$

Clearly, $|\Pi_0| = I < \infty$ because the diameter of $\mathcal{F}_P$ is bounded as stated in Theorem 5.1. We can also verify that $\Pi_0$ satisfies $\sum_{i \in [I]} w_i(\mathbf{p}) = 1$ for all $\mathbf{p} \in \mathcal{F}_P$ because $w_i(\mathbf{p})$ is defined as the normalized tent function.

Now we prove that $\Pi_0$ satisfies the desired property. Since the agent is adaptively calibrated to $\Pi_0$, we have that $\forall i \in [I]$,

$$\frac{n_{[M]}(i)}{M} \left\| \bar{\mathbf{p}}_{[M]}(i) - \mathbf{h} \right\|_2 \leq \sqrt{m} \lim_{M \to \infty} \frac{n_{[M]}(i)}{M} \left\| \bar{\mathbf{p}}_{[M]}(i) - \mathbf{h} \right\|_\infty \leq \sqrt{m} r(M).$$

Now, for $\delta = 3\varepsilon = \frac{3}{4}\varepsilon_0$, let $D_\varepsilon^{(\delta)} \subseteq D_\varepsilon$ be defined as

$$D_\varepsilon^{(\delta)} = \{ x_i \in D_\varepsilon : \|x_i - \mathbf{h}\| \geq \delta \}. \tag{27}$$

Since $|D_\varepsilon^{(\delta)}| \leq |D_\varepsilon| = I < \infty$, taking the sum of calibration error over bins in $D_\varepsilon^{(\delta)}$, we obtain

$$\sum_{x_i \in D_\varepsilon^{(\delta)}} \frac{n_{[M]}(i)}{M} \left\| \bar{\mathbf{p}}_{[M]}(i) - \mathbf{h} \right\|_2 = \frac{1}{M} \sum_{x_i \in D_\varepsilon^{(\delta)}} \left\| \sum_{s \in [M]} w_i(\mathbf{p}_s)(\mathbf{h} - \mathbf{p}_s) \right\|_2 \leq \sqrt{m} I r(M). \tag{28}$$

We can further lower bound (28) and get:

$$\frac{1}{M} \sum_{x_i \in D_\varepsilon^{(\delta)}} \left\| \sum_{s \in [M]} w_i(\mathbf{p}_s)(\mathbf{h} - \mathbf{p}_s) \right\|_2 = \frac{1}{M} \sum_{x_i \in D_\varepsilon^{(\delta)}} \left\| \sum_{s \in [M]} w_i(\mathbf{p}_s)\big((\mathbf{h} - x_i) + (x_i - \mathbf{p}_s)\big) \right\|_2$$

$$\overset{(a)}{\geq} \frac{1}{M} \sum_{x_i \in D_\varepsilon^{(\delta)}} \left( \left\| \sum_{s \in [M]} w_i(\mathbf{p}_s)(\mathbf{h} - x_i) \right\|_2 - \left\| \sum_{s \in [M]} w_i(\mathbf{p}_s)(x_i - \mathbf{p}_s) \right\|_2 \right)$$

$$\overset{(b)}{\geq} \frac{1}{M} \sum_{x_i \in D_\varepsilon^{(\delta)}} \sum_{s \in [M]} w_i(\mathbf{p}_s)\big( \|\mathbf{h} - x_i\|_2 - \|x_i - \mathbf{p}_s\|_2 \big)$$

$$\overset{(c)}{\geq} \frac{1}{M} \sum_{x_i \in D_\varepsilon^{(\delta)}} n_{[M]}(i)(\delta - 2\varepsilon) \geq \frac{\varepsilon_0}{4M} \sum_{x_i \in D_\varepsilon^{(\delta)}} n_{[M]}(i).$$

In the above inequalities, (a) and (b) are both due to triangle inequalities, and (c) is because $\|\mathbf{h} - x_i\|_2 \geq \delta$ from the definition of $D_\varepsilon^{(\delta)}$ in (27) and $\|x_i - \mathbf{p}_s\|_2 < 2\varepsilon$ whenever $w_i(\mathbf{p}_s) > 0 \iff \Lambda(\mathbf{p}_s; x_i, 2\varepsilon) > 0$. Together with (28), the above set of inequalities imply

$$\frac{1}{M} \sum_{x_i \in D_\varepsilon^{(\delta)}} n_{[t]}(i) \leq \left(\frac{4}{\varepsilon_0}\right) \frac{1}{M} \sum_{x_i \in D_\varepsilon^{(\delta)}} \left\| \sum_{s \in [M]} w_i(\mathbf{p}_s)(\mathbf{h} - \mathbf{p}_s) \right\|_2 \leq \frac{4\sqrt{m}I}{\varepsilon_0} r(M). \tag{29}$$

On the other hand, since $D_\varepsilon$ is an $\varepsilon$-grid of $\mathcal{F}_P$, if $\|\mathbf{p}_s - \mathbf{h}\|_2 \geq \varepsilon_0$, there must exist $x_i \in D_\varepsilon$ such that $\|x_i - \mathbf{p}_s\|_2 \leq \varepsilon$, which implies

$$\|x_i - \mathbf{h}\|_2 \geq \|\mathbf{p}_s - \mathbf{h}\|_2 - \|x_i - \mathbf{p}_s\|_2 \geq \varepsilon_0 - \varepsilon = \frac{3}{4}\varepsilon_0 = \delta \quad \Rightarrow \quad x_i \in D_\varepsilon^{(\delta)}.$$

As for the weight that $w_i$ assigns to $\mathbf{p}_s$, we also have

$$w_i(\mathbf{p}_s) = \frac{\Lambda(\mathbf{p}_s; x_i, 2\varepsilon)}{\sum_{j \in [I]} \Lambda(\mathbf{p}_s; x_j, 2\varepsilon)} \geq \frac{2\varepsilon - \varepsilon}{I \cdot 2\varepsilon} = \frac{1}{2I}.$$

Therefore, we have

$$\frac{1}{M}\Big|\{s \in [M] : \|\mathbf{p}_s - \mathbf{h}\| \geq \varepsilon_0\}\Big| \leq \frac{1}{M} \sum_{x_i \in D_\varepsilon^{(\delta)}} \sum_{s \in [M]} (2I) w_i(\mathbf{p}_s) = \frac{2I}{t} \sum_{x_i \in D_\varepsilon^{(\delta)}} n_{[M]}(i) \quad (30)$$

Finally, combining inequalities (29) and (30), we conclude that

$$\frac{1}{M}\Big|\{s \in [M] : \|\mathbf{p}_s - \mathbf{h}\| \geq \varepsilon_0\}\Big| \leq (2I) \lim_{M \to \infty} \frac{1}{M} \sum_{x_i \in D_\varepsilon^{(\delta)}} n_{[M]}(i) \leq \frac{8\sqrt{m}I^2}{\varepsilon_0} r(M),$$

which proves (25). The proof is complete by taking the limit of $M \to 0$, which guarantees $r(M) \to 0$ and immediately implies the convergence result in (24). $\qquad\square$

Note that the rate in Equation (25) does not depend on strategy $\mathbf{h}$. Therefore, in the context of running LAZYGDWOG (Algorithm 3), we can turn Lemma E.1 into the following uniform convergence result across epochs:

**Proposition E.2.** *For any $\varepsilon_0 > 0$, there exists a finite binning $\Pi_0$, such that $\forall \Phi > 0$, if the principal runs* LAZYGDWOG *for $\Phi$ epochs where each epoch has length $M$, and the agent's forecasts $(\mathbf{p}_{\phi,s})_{\phi \in [\Phi], s \in [M]}$ are adaptively calibrated with respect to $\Pi_0$, then we have the following* uniform convergence *guarantee:*

$$\lim_{M \to \infty} \sup_{\phi \in [\Phi]} \frac{1}{M}\Big|\{s \in [M] : \|\mathbf{p}_{\phi,s} - \mathbf{h}_\phi\|_2 \geq \varepsilon_0\}\Big| = 0 \qquad (31)$$

**Remark E.3.** *Note that the rate in (25) has a polynomial dependency on $|\Pi_0|$, which, due to the construction in the proof of Lemma E.1, ends up being exponential in $m$ because it is the size of a $\frac{\varepsilon_0}{4}$ grid of the domain $\mathcal{A}_P$. To improve on this exponential dependency, one possible approach is to design an adaptive calibration algorithm for the agent that achieves the stronger notion of $\ell_1$ calibration, which is more common in recent literature. For example, Hart [36], Foster and Vohra [29, 30] are defined using $\ell_1$ calibration error rather than $\ell_\infty$. Another approach is to avoid using naive conversion from $\ell_\infty$ to $\ell_1$ calibration error in (28), which leads to a polynomial dependency on the number of bins. These two approaches are equivalent ways of formulating the problem, and they both lead to interesting open directions.*

### E.3 More auxiliary lemmas: approximate best response and closeness in utility

In this section, we use the results in Appendix E.2 to show that the average feedback $\frac{1}{M} \sum_{s \in [M]} y_{\phi,s}$ in epoch $\phi \in [\Phi]$ is close to the best response $\text{BR}(\mathbf{h}_\phi)$ (Lemma E.4), and that the principal's average utility in this epoch is close to $U_P(\mathbf{h}_\phi, \text{BR}(\mathbf{h}_\phi))$ (Lemma E.5).

**Lemma E.4.** *For any $\varepsilon_1 > 0$, there exists a finite binning $\Pi_0$ and $M_0 < \infty$ such that when agent's forecasts $\mathbf{p}_{1:t}$ are adaptively calibrated with respect to $\Pi_0$, then we have that $\forall M \geq M_0$,*

$$\sup_{\phi \in [\Phi]} \left\| \frac{1}{M} \sum_{s \in [M]} y_{\phi,s} - \text{BR}(\mathbf{h}_\phi) \right\|_2 \leq \varepsilon_1.$$

*Proof.* Let $\varepsilon_0 = \frac{\varepsilon_1}{2L_{\text{BR}}}$ and $\Pi_0$ be the binning that satisfies Proposition E.2 for parameter $\varepsilon_0$. Therefore, we know from Equation (31) in Proposition E.2 that for $\varepsilon_2 = \frac{\varepsilon_1}{2 \cdot D_P \cdot L_{\text{BR}}}$ there exists $M_0$ such that $\forall M \geq M_0$,

$$\sup_{\phi \in [\Phi]} \frac{1}{M}\Big|\{s \in [M] : \|\mathbf{p}_{\phi,s} - \mathbf{h}_\phi\|_2 \geq \varepsilon_0\}\Big| \leq \varepsilon_2. \qquad (32)$$

Using Lipschitzness of the best response mapping $\mathtt{BR}(\cdot)$, we have that $\forall \phi \in [\Phi]$,

$$\left\| \frac{1}{M} \sum_{s \in [M]} y_{\phi,s} - \mathtt{BR}(\mathbf{h}_\phi) \right\|$$

$$\leq \frac{1}{M} \sum_{s \in [M]} \|y_{\phi,s} - \mathtt{BR}(\mathbf{h}_\phi)\|_2 \qquad \text{(Triangle inequality)}$$

$$\leq L_{\mathtt{BR}} \frac{1}{M} \sum_{s \in [M]} \|\mathbf{p}_{\phi,s} - \mathbf{h}_\phi\|_2 \qquad \text{($\mathtt{BR}(\cdot)$ is $L_{\mathtt{BR}}$-Lipschitz)}$$

$$\leq L_{\mathtt{BR}} \frac{1}{M} \left( \sum_{s \in [M]: \|\mathbf{p}_{\phi,s} - \mathbf{h}_\phi\| \geq \varepsilon_0} \mathtt{diam}(\mathcal{H}_P) + \sum_{s \in [M]: \|\mathbf{p}_{\phi,s} - \mathbf{h}_\phi\| < \varepsilon_0} \varepsilon_0 \right)$$

$$\leq L_{\mathtt{BR}} \frac{1}{M} \left( \varepsilon_2 M \cdot D_P + M \cdot \varepsilon_0 \right) \qquad \text{(Eq. (32) \& $\mathtt{diam}(\mathcal{H}_P) \leq D_P$)}$$

$$\leq D_P \cdot L_{\mathtt{BR}} \cdot \varepsilon_2 + L_{\mathtt{BR}} \cdot \varepsilon_0 = \frac{\varepsilon_1}{2} + \frac{\varepsilon_1}{2} = \varepsilon_1.$$

$\square$

**Lemma E.5.** *For any $\varepsilon_1 > 0$, there exists a finite binning $\Pi_0$ and $M_0 < \infty$ such that when agent's forecasts $\mathbf{p}_{1:t}$ are adaptively calibrated with respect to $\Pi_0$, then we have that $\forall M \geq M_0$,*

$$\sup_{\phi \in [\Phi]} \left| \frac{1}{M} \sum_{s \in [M]} U_P(\mathbf{h}_\phi, y_{\phi,s}) - U_P(\mathbf{h}_\phi, \mathtt{BR}(\mathbf{h}_\phi)) \right| \leq \varepsilon_1. \tag{33}$$

*Proof.* The proof of this lemma is very similar to that of Lemma E.4, with a different choice of constants $\varepsilon_0$ and $\varepsilon_2$. Note that since $U_P$ is $L_2$-Lipschitz in the second argument, we have

$$\left| \frac{1}{M} \sum_{s \in [M]} U_P(\mathbf{h}_\phi, y_{\phi,s}) - U_P(\mathbf{h}_\phi, \mathtt{BR}(\mathbf{h}_\phi)) \right| \leq \frac{1}{M} \sum_{s \in [M]} \|U_P(\mathbf{h}_\phi, y_{\phi,s}) - U_P(\mathbf{h}_\phi, \mathtt{BR}(\mathbf{h}_\phi))\|_2$$

$$\leq L_2 \cdot \frac{1}{M} \sum_{s \in [M]} \|y_{\phi,s} - \mathtt{BR}(\mathbf{h}_\phi)\|_2.$$

The rest of the proof follows from Lemma E.4 by choosing $\varepsilon_0 = \frac{\varepsilon_1}{2 L_2 L_{\mathtt{BR}}}$ and $\varepsilon_2 = \frac{\varepsilon_1}{2 \cdot D_P \cdot L_{\mathtt{BR}} L_2}$. $\square$

## F  Future Directions

**Adaptive Calibration**    Although the results in this paper are all stated in terms of the $\ell_\infty$-calibration error (e.g., *maximum* instead of *sum* over bins), a lot of the existing calibration literature focuses on $\ell_1$-calibration error [29, 30]. It is an interesting problem whether we can get $\ell_1$-adaptive calibration error bounds without a polynomial dependency in the number of binning functions, where obtaining such bounds lead to polynomial improvements on the dependency of $m$ (the number of agent's actions). In the case of continuous calibration, it is an open problem to obtain uniform (adaptive) calibration error bounds for parametric or nonparametric continuous binning function classes. Resolving this open problem could lead to a better rate for the learning direction of Theorem 5.2, as the current result uses naive $\ell_\infty$-to-$\ell_1$ conversion of calibration error that leads to linear dependency on the number of binning functions, which turns out to be exponential in the dimension of the principal's action space. See Remark E.3 for more details.

**Convex optimization from membership oracles**    In the case of finite Stackelberg games, we use the results of Lee et al. [42] to solve the constrained convex optimization problem in each polytope from approximate membership queries, where the key step is a polynomial reduction from approximate separation oracles to approximate membership oracles. However, the precision of the constructed separation oracle is worse than the precision of the membership oracle, which naturally leads to a

worse precision of the final optimization solution. To be more specific, given an $\varepsilon$-membership oracle to a convex set $K$, Lee et al. [42] can only guarantee the returned solution to be $\varepsilon^{1/\gamma_0}$-optimal and contained in $B_2(K, \varepsilon^{1/\gamma_0})$, where $\gamma_0 > 6$ is a fixed constant. An understudied open direction in the optimization community is whether a variant of Lee et al. [42] can return a solution in $B_2(K, \varepsilon + \delta)$ for $\delta$ that is *any* tunable parameter, with runtime depending polynomially on $1/\delta$.

**Open Problem F.1.** *Given an $\varepsilon$-approximate membership oracle of a convex set $K \subseteq \mathbb{R}^m$ and an evaluation oracle of a convex function $f$, is there an optimization algorithm that, under sufficient regularity conditions, for any tunable $\delta$, finds a near-optimal solution $\hat{x}$ such that $f(\hat{x}) \leq \min_{x \in K} f(x) + \varepsilon^{1/\gamma_0}$ and $\hat{x} \in B_2(K, \varepsilon + \delta)$ within $poly(m, 1/\delta, \log(1/\varepsilon))$ calls to both oracles?*

We note that resolving Open Problem F.1 will immediately lead to exponentially improved rates for our Theorem C.14 for any finite Stackelberg games. To see this, recall that the algorithms that achieve Theorem C.14 require repeated calls to the constructed APPROXMEM oracle. This oracle provides an $\varepsilon_1$-approximate response, indicating whether or not the query point belongs to the $\varepsilon_2$-conservative version of each best response polytope. The number of samples used in simulating each oracle call is exponential in $\frac{\varepsilon_2}{\varepsilon_1}$. The algorithm by Lee et al. [42] amplifies the inaccuracy of APPROXMEM at a polynomial rate of $\gamma_0$. Consequently, imposing a constraint of $\varepsilon_1 \lesssim \varepsilon_2^{\gamma_0}$. However, resolving Open Problem F.1 removes this constraint and introduces a much milder one, i.e., $\varepsilon_1 \lesssim \varepsilon_2 - \delta$, where $\delta$ is a tunable parameter. This adjustment leads to exponential improvement in the final sample complexity.

