# OpenReview forum: "Calibrated Stackelberg Games: Learning Optimal Commitments Against Calibrated Agents"
_NeurIPS.cc/2023/Conference — NeurIPS 2023 spotlight_

### Official Review · Reviewer_j8Y8 · 2023-07-05

**Soundness:** 4 excellent
**Presentation:** 4 excellent
**Contribution:** 3 good
**Rating:** 7
**Confidence:** 3

**Summary:**

This manuscript introduces the concept of calibrated forecasts in repeated Stackelberg games (SG) and proposes two concepts: the calibrated Stackelberg games (CSG) that generalizes the standard SGs and the adaptive calibrated forecast. The technical contribution is as follows: First, a principal's learning algorithm against adaptively calibrated agents is proposed where the average utility of the principal converges to the Stackelberg value, which is the best possible average utility under this setting. Second, a forecasting algorithm that meets the concept of calibrated forecast is proposed, which means that the agents can actually perform calibrated forecasts for the principal's action. Third, even for continuous Stackelberg games where the action sets of principal and agent are continuous, there will be principal's learning algorithm against certainly adaptively calibrated agents where the average utility of the principal converges to the Stackelberg value, the best possible one.

**Strengths:**

The notion of calibrated forecast, on which the paper is entirely based, seems much reasonable; compared to the original definition, it is generalized by introducing a binning function, enabling us to represent some rules for choosing the best response of agents (deterministic/randomized tie-breaking rule in the manuscript). Although its assumption seems much stronger and unrealistic, this work succeeded in proposing forecasting algorithms that meets the conditions of adaptive calibration by combining online algorithms with the study on novel game dynamics; its technical contribution seems far non-trivial. For the principal, this work succeed in proposing learning algorithms that achieves the best possible average utility asymptotically, meaning that the proposed learning algorithm is asymptotically the best one under this setting.

**Weaknesses:**

The connection between the proposed concepts (calibrated Stackelberg games and adaptive calibrated forecast) and the applications the manuscript claims (Stackelberg security games and strategic classification) is unclear from the manuscript although it is claimed that the results "immediately apply". Thus, it looks like that this work addresses an artificial setting in Stackelberg games. To avoid this, the authors should carefully review the connection between the proposed concepts and the applications at least in the Appendix.

Minor comment:
In Theorem 5.2, unlike Theorem 3.1, the binning \Pi_0 is fixed as Eq. (25), but it is described only in the Appendix. As far as I read the main part, I'm afraid that the binning is irresponsible and not related to the agent's policy. For the sake of completeness, please consider describing the actual formula for the binning and its meaning in the main article.

**Questions:**

P.3, l.128: "Definition 2.2 is weaker than the standard definition of calibration..." What does "weaker" mean? As far as I understand, introducing a binning function \Pi is a generalization compared to the standard definition but it does not strengthen or weaken the assumption.

**Limitations:**

The authors describe some limitations (and thus future directions) of this work in the conclusion section and in Appendix F.

---

> ### Author Rebuttal · Authors · 2023-08-09
>
> We appreciate the reviewer's positive feedback on our paper. We’re happy to see that the reviewer recognizes the importance of addressing Stackelberg games through calibration, moving beyond traditional assumptions. We address the specific questions in the subsequent paragraphs:
>
> **Stackelberg security games and strategic classification as applications of CSG**
>
> Thank you for the suggestion. We'll add the following paragraph to the revised version of our paper to explain how security games and strategic classification falls under the framework of CSGs:
> - **Stackelberg security games (SSG)** are a prominent example of finite Stackelberg games that captures the strategic interactions between an attacker (or agent) and a defender (or principal). In an SSG, the defender commits to a probabilistic distribution of security resources across $k$ targets, and the attacker best responds to it by attacking the target that maximizes its utility. Specifically, the defender's action space $\mathcal{A}_P$ includes the finite set of schedules, where each schedule is the subset of targets that can be simultaneously defended by one security resource. The attacker's action space $\mathcal{A}_A$ is the finite set of targets $[k]$ that the agent can attack. Transitioning this into our Calibrated Stackelberg Game (CSG) model, we depart from the traditional assumption that the attacker has perfect knowledge of the defender's strategy. Instead, we consider realistic attackers that base their decisions on calibrated forecasts of defender's strategies.
> - **Strategic classification** is an application of continuous SGs where the principal aims to learn classifiers that remain robust when agents strategically manipulate their features to receive positive classifications. We use the model of [Dong et al, EC 2018] to illustrate how it falls under the framework of continuous SGs. In this model, during every round of interaction, the principal commits to a linear classifier $\mathbf{h}_t\in\mathbb{R}^d$, then the agent with initial feature $\theta_t\in\mathbb{R}^d$ best responds to the classifier $\mathbf{h}_t$ by modifying its feature to $\tilde{\theta}_t=\theta_t+w_t\in\mathbb{R}^d$ that maximizes the dot product of $\langle \tilde{\theta}_t,\mathbf{h}_t\rangle$ minus the cost of movement $c_t(w_t)$. The principal's utility is the negative logistic loss or hinge loss. In the above model, the agent has full knowledge of the classifier $\mathbf{h}_t$ before manipulating its features, which might be unrealistic for applications like college admissions where the classification rules are opaque to the agent. Our CSG framework relaxes this assumption by allowing agents to best respond to beliefs about the classifier $\mathbf{h}_t$ generated by any calibrated forecasting algorithm. Our results applied to this setting show that when agents’ features are drawn from a stochastic distribution, the principal's optimal average utility is captured by the same Stackelberg value obtained when agents have direct knowledge of $\mathbf{h}_t$.
>
> **Binning function in the continuous setting**
>
> While the continuous setting’s binning may appear less intuitive, it is still related to the agent’s policy. In the discrete setting, the grouping of the principal's strategies into bins is based on whether they induce the same best response from the agent. When extending this to a continuous setting, the concept remains grounded in the agent's best response, but with some adjustments to accommodate the continuous action space. Specifically, we define the binning function based on whether the induced best responses are in close proximity rather than being identical. This is achieved by first creating an $\varepsilon$-net of the strategy space, followed by smoothing the discontinuous net into continuous $\Lambda$ functions that serve as the bins after normalization. Intuitively, the $\Lambda$ functions, defined as $\Lambda(p)=(R-\\|p-x\\|_2)\_+$ for all $x$ in the net, resemble the shape of "tents". These functions peak at $x$ and smoothly decrease to 0 as $p$ moves away from $x$. Consequently, strategies sharing a nonzero $\Lambda$ function (or equivalently nonzero binning function) must be close in $l_2$ distance. This, in turn, implies that their best responses must also be close, given the assumption that the BR function is Lipschitz. In this manner, the cover still respects the distance metric regarding the agent's response, ensuring that the binning is not irresponsible or unrelated to the agent's policy, but rather a nuanced adaptation to the continuous setting. We will further clarify this in the final version of the paper.
>
>
> **Binning as a weakening of the standard calibration assumptions**
>
> While the introduction of binning is indeed a generalization of the standard calibration definition, it's a definition that can only weaken the assumption, because binnings serve as a coarsening of the representation of functions on which the agent wants to achieve vanishing calibration error.
>
> In fact, the finest-grain bin, which corresponds to the indicator function $w_{\mathbf{p}}(x)=\mathbf{1}\\{x=\mathbf{p}\\}$ for every possible strategy $\mathbf{p}$, would lead to the standard definition of calibration as it inspects the calibration error for every strategy independently; and calibration wrt this binning immediately imply calibration wrt all other normalized binnings. Given our approach of defining the binning based on the agent's best response function, our generalized calibration assumption is weaker than the finest-grained binning and forms a realistic assumption that the agents can easily achieve.

---

> > ### Comment · Reviewer_j8Y8 · 2023-08-14
> >
> > Thank you very much for detailed reply. I adequately understand the connection between the proposed concepts and the applications (SSG and strategic classification). I think the authors' claim that the proposed concepts fit the described applications is appropriate. In addition, the question I raised is adequately resolved. Thus, I'm still in favor of accepting this manuscript.

---

> > > ### Author Response · Authors · 2023-08-14
> > >
> > > Thank you for taking the time to read our rebuttal. We are glad that our response addressed your questions. We'll use the additional page to review the connections between our proposed CSG framework and relevant applications.

---

### Official Review · Reviewer_M2eA · 2023-07-05

**Soundness:** 3 good
**Presentation:** 2 fair
**Contribution:** 3 good
**Rating:** 6
**Confidence:** 3

**Summary:**

In this work, the authors consider a problem of Calibrated Stackelberg Games (CSG), which is a generalization of the Stackelberg Games. These framework differ from the standard online learning problems as in the SG framework instead of only having a single learner entity, there is a principal and an agent. The key difficulty introduced by the CSG framework is that the agent needs to respond to the action of the principal without being able to observe it. Instead, the agent have to forecast what they expect the principal's action to be and aim to best respond to that believed action from the principal.
As this learning objective is more challenging than classic learning problems, the authors consider the question of whether the principal can achieve optimal utility (V* being the utility of the principal's action with the highest best response) in this calibrated game.
An important part of the paper is dedicated to the construction of adaptively calibrated forecasts and of the CSG protocol. Then, the authors present and analyse an algorithm that can achieve optimal utility with high probability.
The principal's algorithm is a simple explore then commit strategy. This exploration starts with  $log T/\eta$ uniform uniform strategies samples, for which the agent returns an associated response. Then the principal tries to find approximately optimal strategies for each of the agent's response, and ends up picking the one of these that yields the highest utility.
They then analyze the performance of the algorithm and show that it assumptotically reaches the optimal utility for discrete and continuous games.

**Strengths:**

This paper studies a generalization of the Stackelberg games, which are challenging in the online learning framework, and show that it is possible to reach optimal utility for the principal even if the agent doesn't have access to the strategy picked by the principal ahead of time.
These results are novel, and the analysis of the algorithm builds upon standard online learning algorithms. The authors took particular care in connecting the CSG framework with other online learning problems such as sleeping experts.

This work provides some good preliminary baselines for the SCG problem, and should provide a strong foundation for future works to build upon it.

**Weaknesses:**

The main weakness of the paper is that the results provided, meaning that the algorithm presented can find the optimal utility, only holds asymptotically, making it difficult to appear useful in practice, when we only have a limited time-horizon.
It would be useful to discuss extensions of this work that could achieve stronger guarantees in finite time horizons.


**Questions:**

Do you think that it is possible to extend your work beyond the asymptotic guarantees that you provide?

---

> ### Author Rebuttal · Authors · 2023-08-09
>
> We thank the reviewer for the appreciation of our work.
>
> **On whether Nonasymptotic guarantees are possible**
>
> Yes, absolutely! While we state our results in asymptotic forms in the main body, we have already provided non-asymptotic guarantees for finite time horizon in the appendix. Specifically, please refer to Thm C.14 in Appendix C.6 for a more formal version of Thm 3.2. We opted to defer these detailed rates to the appendix due to their complexity and the potential difficulty in interpretation for general games, but we are happy to instantiate them for specific settings like security games and present them as corollaries in the main body.

---

> > ### Comment · Reviewer_M2eA · 2023-08-16
> >
> > Thank you for your answer, I had indeed overlooked that part of the theorem.

---

### Official Review · Reviewer_Crqu · 2023-07-07

**Soundness:** 3 good
**Presentation:** 2 fair
**Contribution:** 3 good
**Rating:** 7
**Confidence:** 3

**Summary:**

The paper defines and studies a new Stackelberg games setup. Rather than making some standard assumptions --- e.g. that the principal and/or the agent exhibit specific types of play (e.g. agent playing no regret), or assuming access to the agent’s best response oracle, etc. --- this paper only assumes that the agent will be playing by best-responding to (appropriately) calibrated forecasts of the principal play. In this Calibrated Stackelberg Games setup, the authors show that: (1) The principal, by only knowing that the agent will be playing in a calibrated way, can achieve exactly (no more and no less than) the Stackelberg value of the game over a repeated interaction, by following an explore-then-commit style algorithm; (2) The agent has an efficient algorithm for producing said calibrated forecasts (in fact, strengthened by the notion of adaptivity --- meaning that calibration should hold over all subintervals of the time axis). These results are further complemented by e.g. considering both finite and continuous action spaces, as well as an adaptive calibration algorithm.

**Strengths:**

As intended by the authors, they are able to demonstrate that Calibrated Stackelberg games indeed show potential towards relaxing/moving away from various restrictive assumptions (on what the principal and the agent observe and how they play) in the literature. An important moral takeaway is that the same old Stackelberg value --- which one might have expected might in fact require the principal and the agent to (more or less) explicitly observe each other’s play --- can be achieved over time by only asking the agent to use calibrated forecasts of the principal’s play (which is a fairly mild requirement), and the principal to know and use the fact that the agent exhibits calibration.

In fact, this conclusion is carefully shown to hold (existentially and algorithmically) both for finite and continuous action spaces --- which, while to be morally expected from standard minimax reasoning, still takes work to rigorously establish and shows thoroughness on the authors’ part.

In terms of the techniques employed, the exploration process for achieving this value on the principal’s side is, in fact, not very straightforward to design both for finite and continuous action spaces; so, there is a healthy dose of sophistication involved. The adaptive calibration algorithm, on the other hand, quite straightforwardly follows from existing techniques at the intersection of no-regret dynamics and online multiobjective settings.


**Weaknesses:**

I did not spot any technical weaknesses, and the contribution of this paper to both the Stackelberg games and the calibration literatures is solid. So overall, the paper does not have any significant weaknesses. However,

I would, however, like to point out that the principal needing, in certain parts of this new framework, to know the calibration rate of the agent is not to be taken lightly given that an important motivation of this paper is to relax, as much as possible, the prevalent specificity, in existing literature, of the requirements on what the principal and the agent should know. I would appreciate it if the authors could provide further elaboration of this, beyond the brief mention in the conclusion.

Secondly, while having *adaptive* calibration guarantees is nice, compared to regular marginal ones, I’m not clear on how or whether this adaptivity interacts with the proposed theory of CSG or is more or less orthogonal? In other words --- okay, using the standard sleeping experts technique for establishing adaptive online guarantees, it is possible to make the agent calibrated on every [s, t] rather than just on [1, T]; but does that really matter for e.g. being able to achieve the value V* in the process of play, or any other desirable Stackelberg properties? Since the main point of the paper is to propose a theory of calibrated Stackelberg games, it is important to be clear on whether this is an essential element of such a theory or was just added to the paper for good measure. If this is in fact an essential element, I’d like to ask the authors to clarify this.

I also did find the presentation suboptimal --- the paper reads quite densely; especially, in my experience, when it comes to the proof sketch after Theorem 3.2 in Section 3. I invite the authors to revamp that part of the presentation for the rebuttal phase. Some specific things that I’d appreciate would be (1) alleviating the notational/explanatory tedium related to condition (P1) --- I am still not clear on how strong or weak it is, and where exactly things must break if it wasn’t required; (2) adding a high-level description (preferably involving more prose) of how the agent’s calibration figures into what the algorithm for the principal does.


**Questions:**

For the substantive questions, please see my questions above on: (1) the principal needing to know the calibration rate; (2) the requirement of adaptivity of calibration; (3) some elements of Section 3 such as property (P1) etc.

Here, I’ll quickly list a sampling of a few typos and notational issues:
Line 300: bound*ed*
Line 285: Where is the notation L_g defined?
Line 276: Brackets around sigma in the subscript for g
Line 222: Probably meant to say that (h hat, y) is an equilibrium rather than just h hat
Line 201: Where is h bar_T defined?
Line 174: The fundamental constructs from Definition 2.2 are reviewed, not from Eq 1
Line 102: converged *to a* Stackelberg equilibrium

---

> ### Author Rebuttal · Authors · 2023-08-09
>
> We thank the reviewer for their positive and insightful comments. Below we address the specific questions.
>
> **The principal needs to know the agent’s calibration rate**
>
> We will add a more detailed discussion to our paper.
>
> On the one hand, the principal does *not* need to know the agent's exact calibration rate. Knowing an upper bound or approximate value suffices for learning a near-optimal commitment against calibrated agents. Given that the calibration error has to be small and vanishing for our setting, we think that assuming some knowledge of an appropriate upper bound on a small calibration error provides a good tradeoff between relaxing the assumptions found in previous literature and still obtaining provable guarantees in Calibrated Stackelberg games.
>
> On the other hand, if the principal is completely unaware of the calibration rate or the calibration error is large, several complications arise. Since the agent's beliefs might come from any forecasting algorithm, their (average) responses to the principal's strategies may be suboptimal. This uncertainty complicates the exploration phase, where the principal must decide between improving feedback accuracy by repeating the current strategy or updating the strategy based on existing feedback. Therefore, we believe that some degree of knowledge regarding calibration rate is necessary, and proving this necessity is an intriguing question for future research. Moreover, if the calibration error is large (and thus harder to justify having an approximate upper bound), the agent’s behavioral model can fit any online algorithm. It is known that in such situations, the principal’s utility is not characterized by $V^*$. For example, during a commit phase, even playing the optimal strategy $h^*$ may lead to high suboptimality if the agent doesn't best respond due to high calibration error of the forecasts.
>
> **The requirement of adaptivity**
>
> We believe that lack of adaptive guarantees poses a technical challenge for the design of learning algorithms and likely impacts the convergence rates to $V^*$. At a high level, adaptivity ensures that each exploration step (referring to a sub-interval) is approximately correct, enabling the principal to more effectively search the space and find a final strategy that is both robust and near-optimal.
>
> More specifically, recall that during the exploration phase of the Explore-Then-Commit algorithm, the principal first learns the optimal commitments within each robustified best response polytope, then selects the one with the highest utility. In this process, the key challenge is determining whether a strategy robustly lies within a specific best response polytope. We address this by sampling test strategies around the queried one, playing them repeatedly to find an approximate best response. This is enabled by adaptive calibration within time subintervals $[t_1,t_2]$ where each test strategy is played. Without the adaptivity property, however, ensuring an approximate best response would require a much larger $t_2$ relative to $t_1$ to make the interval $[0,t_1)$ negligible compared to $[0,t_2]$, where marginal calibration applies. This increases the repetitions for each test strategy. While this is not a proof of an algorithm-independent lower bound on the convergence rate, we wouldn’t be surprised if such a proof could be formalized that indicates that the lack of adaptivity property would result in the convergence rate becoming much worse, possibly exponentially. We believe this is an interesting question for future work.
>
> **The presentation in Section 3**
>
> We plan to enhance the clarity and structure of Section 3.1 as follows:
> - **Elaborating on the objectives for Explore phase**: Before delving into the notations $B\_2(S,\varepsilon)$ and $B\_2(S,-\varepsilon)$, we'll overview what we aim to achieve in the Explore phase:
>   - **Idealized Setting**: Initially, we'll consider a setting with zero calibration error, where the agent's forecasting algorithm is perfectly and adaptively calibrated, leading to $y\_t=\text{BR}(h_t)$ at every round. Within the Explore phase, the task simplifies to identifying a near-optimal strategy through best response oracles, satisfying $U_P(\tilde{h},\text{BR}(\tilde{h}))\ge V^*-\varepsilon\_1$ for a predetermined $\varepsilon\_1$. In the Commit phase, given that the agent always plays $\tilde{y}=\text{BR}(\tilde{h})$, the Stackelberg regret can be upper bounded by $\varepsilon\_1|T\_2|$. Hence, the Explore-Then-Commit algorithm's regret is bounded by $V^*|T_1|+\varepsilon_1|T_2|$.
>   - **Realistic Setting**: Moving away from the idealized setting, we must account for possible discrepancies between $y\_t$ and $\text{BR}(h_t)$ due to calibration error. This introduces: (1) An increased sample complexity in the Explore phase, given the necessity to learn a near-optimal strategy from noisy responses; (2) Potential deviations from the action $\tilde{y}=\text{BR}(\tilde{h})$ due to miscalibrations in belief. To address the first challenge, we employ Algorithm 2 which constructs an *approximate* best response oracle by repeatedly interacting with a calibrated agent. For the second challenge, we require our learned policy $\tilde{h}$ to be robust against inaccurate forecasts. This is reflected in condition (P1), which necessitates the ball of radius $\varepsilon_2$ around $\tilde{h}$ to be fully contained in the polytope $P_{\tilde{y}}$. The critical insight from (P1) is: for any forecast $p_t$ that results in a best response $y_t\neq\tilde{y}$, there must be a minimum distance of $\varepsilon\_2$ separating $p_t$ from $\tilde{h}$. Combined with the definition of calibration error, this relationship allows us to establish an upper bound on the number of such rounds.
>
> - **Added figure**: We'll add two figures illustrating the notations $B\_2(S,\pm\varepsilon)$ and the relation between $\tilde{h}$ and $\bar{p}_{T_2}$. The PDF containing the figures is attached to the global response.

---

> > ### Comment · Reviewer_Crqu · 2023-08-16
> > **Acknowledgment**
> >
> > Thank you to the authors for providing a detailed and informative response to my 3 main questions.
> >
> > The point regarding the potential necessity of adaptivity for obtaining good/improved convergence rates is interesting, and I now agree that adaptivity fits into the scope of the manuscript sufficiently naturally.
> >
> > Also, the reworked paragraph on the specifics of the algorithmic contribution in Section 3.1 is much appreciated, and the attached graphics are clean and informative.
> >
> > Therefore, I've increased my score for the paper and maintain my positive opinion of it.

---

> > > ### Author Response · Authors · 2023-08-17
> > >
> > > Thank you for taking the time to read our rebuttal and for increasing the score!

---

### Author Rebuttal · Authors · 2023-08-09

Please refer to the attached PDF for the added figures.

---

### Decision · Program_Chairs · 2023-09-21

**Decision:**

Accept (spotlight)

**Comment:**

This paper introduces a new framework---Calibrated Stackelberg Games, which is a generalization of the standard Stackelberg Games (SGs) framework. Rather than standard strong assumptions on the behaviors of principle and agents, this new framework only assumes that the agent will be playing by best-responding to calibrated forecasts of the principal play. This paper further consider a stronger notion of calibration---adaptive calibration, and show that the principal can achieve utility that converges to the optimum Stackelberg value of the game.

The setting is novel which addresses certain drawbacks of prior framework, and the results make solid contribution to the community. The author feedback effectively addressed the concerns of reviewers. We thus recommend acceptance.